# Focused ultrasound enables selective actuation and Newton-level force output of untethered soft robots

Bo Hao[1,7], Xin Wang[1,7], Yue Dong [2,7] ✉, Mengmeng Sun[1], Chen Xin[1], Haojin Yang[1], Yanfei Cao[1], Jiaqi Zhu[1], Xurui Liu[1], Chong Zhang[1], Lin Su[1], Bing Li [2] ✉ & Li Zhang [1,3,4,5,6] ✉

Untethered miniature soft robots have significant application potentials in biomedical and industrial fields due to their space accessibility and safe human interaction. However, the lack of selective and forceful actuation is still challenging in revolutionizing and unleashing their versatility. Here, we propose a focused ultrasound-controlled phase transition strategy for achieving millimeter-level spatially selective actuation and Newton-level force of soft robots, which harnesses ultrasound-induced heating to trigger the phase transition inside the robot, enabling powerful actuation through inflation. The millimeter-level spatial resolution empowers single robot to perform multiple tasks according to specific requirements. As a concept-of-demonstration, we designed soft robot for liquid cargo delivery and biopsy robot for tissue acquisition and patching. Additionally, an autonomous control system is integrated with ultrasound imaging to enable automatic acoustic field alignment and control. The proposed method advances the spatiotemporal response capability of untethered miniature soft robots, holding promise for broadening their versatility and adaptability.

Untethered soft miniature robots have demonstrated significant potential in various applications including manipulation[1,2], sensing[3], on-demand drug delivery[4,5], and minimally invasive surgery[6,7]. The inherent advantages of those robots, such as safe human-machine interaction[8], environmental adaptability[9], and the ability to access narrow spaces within the body[10–13], have attracted increasing interest in promoting their clinical and industrial translation[14,15]. However, a trade-off exists within current untethered systems, i.e., whilst their wireless nature improves accessibility compared to tethered-actuated robots, particularly in hard-to-reach regions of the body[16–19], it concurrently

compromises their ability for reliable and accurate actuation[20] and limits the force output[21]. Thus, the biomedical potential and practical application capabilities of untethered soft robots are still far from their tethered counterparts. Developing a new actuation method for untethered miniature soft robots that satisfy the requirements of precise control in deep tissue or liquid environments for the necessary multi-task performing ability, and can achieve significant force output, remains challenging.

One of the primary challenges faced by untethered soft robots is the need for an effective selective actuation method in confined space

[1]Department of Mechanical and Automation Engineering, The Chinese University of Hong Kong, Hong Kong SAR 999077, PR China. [2]Guangdong Provincial Key Laboratory of Intelligent Morphing Mechanisms and Adaptive Robotics, School of Mechanical Engineering and Automation, Harbin Institute of Technology, Shenzhen 518055, PR China. [3]Multi-Scale Medical Robotics Center, Hong Kong Science Park, Shatin NT, Hong Kong SAR 999077, PR China. [4]CUHK T Stone Robotics Institute, The Chinese University of Hong Kong, Hong Kong SAR 999077, PR China. [5]Chow Yuk Ho Technology Center for Innovative Medicine, The Chinese University of Hong Kong, Hong Kong SAR 999077, PR China. [6]Department of Surgery, The Chinese University of Hong Kong, Hong Kong SAR 999077, PR China. [7]These authors contributed equally: Bo Hao, Xin Wang, Yue Dong. ✉e-mail: dongyue@hit.edu.cn; libing.sgs@hit.edu.cn; lizhang@cuhk.edu.hk

and deep tissue, allowing one robot to perform multiple functions depending on the task requirements[22]. This is crucial for various applications[9,23], such as minimally invasive surgery, where selective utilization of different tools (e.g., forceps or needle for biopsy and electrodes for coagulation) according to specific tasks[24] is required. However, selective actuation for untethered soft robots is hard to achieve due to the absence of wires or channels. While some advancements have been made in selective actuation, these approaches involve integrating different materials or mechanisms that show distinct responses to specific physical fields, such as light of different wavelengths[25] or magnetic fields of different rotational directions[26]. To avoid increasing the material and structural complexity of the robot, a more straightforward strategy is to leverage non-homogeneous physical fields for spatial selective stimulation, such as light[27] or microwave[28]. However, the lack of methods capable of achieving spatial selective actuation within biological tissues still poses a challenge for the biomedical applications of untethered miniature soft robots. In this regard, focused ultrasound stands out due to its inherent ability to penetrate deep tissue (up to tens of centimeters[29]) and its high resolution (millimeter-level[30]) in tissue/water environments, making it promising for spatially selective actuation of untethered soft robots.

Another challenge for untethered soft robots is the need to generate large force output[21], as numerous biomedical applications require Newton-level forces, including procedures like biopsy (0.3–4 N)[31], stent deployment (over 1 N)[32], or suturing (2–6 N)[6]. However, most untethered soft miniature robots fall short of generating such forces due to the limited power supply of untethered actuation methods and the high energy dissipation characteristics of the constituent soft materials[33,34]. Considering the requirement of these applications, wireless fluidic actuators/robots based on phase transition with considerable force output show great potential to break these restrictions. This is typically accomplished by controlling pressure change inside the soft actuators/robots body by triggering the phase transition of enclosed liquid by external energy[32,35–37], resulting the significant deformation and force output. Additionally, this method does not rely on batteries, making the actuator completely soft and magnetic resonance imaging (MRI) compatible.

Here, we propose a focused ultrasound-controlled phase transition (FUPT) method for the wireless actuation of soft robots, which utilizes the pressure change inside the soft actuator/robot that is triggered by the acoustothermal effect of iron oxide nanoparticles ($Fe_3O_4$ NPs) doped elastomer and the phase transition of enclosed low boiling point liquid to realize controllable deformation/locomotion. The FUPT method not only enables millimeter-level spatial selective actuation for unleashing the versatility of the untethered soft robots, but also allows the designed soft robot to exert a high output force (~5.5 N), satisfying the Newton-level output force requirements for most minimally invasive operations. The tissue and liquid penetration capability of ultrasound also ensures that this approach is suitable for biomedical applications. To prove the concept, we demonstrated an in-pipe robot for locomotion and on-demand liquid cargo delivery, and a biopsy robot capable of carrying out sequential tasks such as needle insertion, tissue acquisition, and tissue patching. Furthermore, we developed a control system that enables automatic acoustic field alignment and control of the focused ultrasound waves based on the observation of robot's state, which makes the FUPT actuation method reported in the current study attractive for various applications.

## Results
### Mechanism of focused ultrasound-controlled phase transition (FUPT) method
The mechanism of the focused ultrasound-controlled phase transition (FUPT) method relies on the energy transformation resulting from the pressure change induced by the phase transition of the low boiling point liquid inside the robot body. As shown in Fig. 1a, the FUPT-based

soft robots/actuators are composed of $Fe_3O_4$ NPs doped Ecoflex 00-30 and injected with the low boiling point liquid (i.e., Novec 7000 engineering fluid, which has a boiling point of 34 °C at 1 atm and exhibits excellent biocompatibility)[38]. A piezoelectric transducers array is utilized to generate focused ultrasound waves, which in turn produce thermal energy within the soft actuator's body based on the acoustothermal effect. The localized temperature rise induces the low boiling point liquid inside the soft actuators/robots to vaporize and increase internal pressure. By appropriately designing the structure, the pressure change is converted into large mechanical output in terms of deformation and force, which allows for controllable locomotion and specific task execution of the soft robot (Fig. 1a). The FUPT actuation method enables the designed soft robot charactered with tissue penetration, selective actuation, and large mechanical output abilities, we demonstrate these through a series of soft robots/actuator with different functionalities. Specifically, benefiting from the tissue penetration ability of ultrasound[39], we have designed a soft capsule that can be remotely triggered through tissue to release the loaded liquid cargo (Fig. 1b). Additionally, as shown in Fig. 1c, the high resolution of the focused ultrasound endows the FUPT based soft robots with selective actuation ability, the multiunit capsule and in-pipe soft robot are demonstrated for the selectively release and on-demand liquid cargo delivery, respectively. Moreover, the combination of the phase transition and structural design induces substantial mechanical output, we demonstrated this through a soft robot with large deformation and force output, that is capable of tissue sample acquisition through needle insertion and secure attachment of wound healing patches in the intestine (Fig. 1d). These features make the FUPT based miniature soft robots/actuators step further for the biomedical applications (Supplementary Table 3 compared FUPT method with some other actuation methods). In the following sections, we present detailed results and typical functional soft devices to illustrate the application potential.

### Influence factors of FUPT actuation method
We investigate the influence factors of the spatial resolution and the acoustothermal induced phase transition. As shown in Fig. 2a, the ultrasound transducer array is fabricated by arranging the PZT transducers distributed on the surface of a sphere with a diameter of 90 mm. Each transducer is oriented towards the center of the sphere. Simulation results illustrate that each transducer contributes to the acoustic wave passing through the center of the sphere, resulting in an increase in intensity and resolution at the focal point. The transducer was numerically simulated and experimentally calibrated, details can be found in Supplementary Section 1, where the relationship between the acoustic pressure amplitude and the applied peak-to-peak voltage ($V_{pp}$) was measured to be 8.16 kPa/$V_{pp}$.

For comparison, Fig. 2b, c depict the calculated distributions of the acoustic field for a single transducer and a transducer array, respectively. As shown in Fig. 2b, the shaded area represents the lateral resolution, defined as the full-width at half maximum (FWHM) intensity resolution. The lateral resolutions for the single transducer and the transducer array are approximately 2.5 mm and 2.6 mm, respectively. Although the transducer array has a slightly lower lateral resolution, it exhibits a 7-fold increase in pressure amplitude compared to the single transducer configuration (Fig. 2b), resulting in higher energy production. Figure 2c compares the longitudinal resolution, revealing a value of 5 mm for the transducer array. In contrast, the single transducer lacks focusing capability in the longitudinal direction. These results indicate that the utilization of the transducer array effectively achieves the focusing of ultrasound waves, thus improving the spatial resolution of the designed acoustic field. In comparison, electromagnetic waves-based methods with similar tissue penetration ability have more than meter-level wavelengths[40], resulting in a resolution significantly lower than that of the FUPT method based on the diffraction limit.

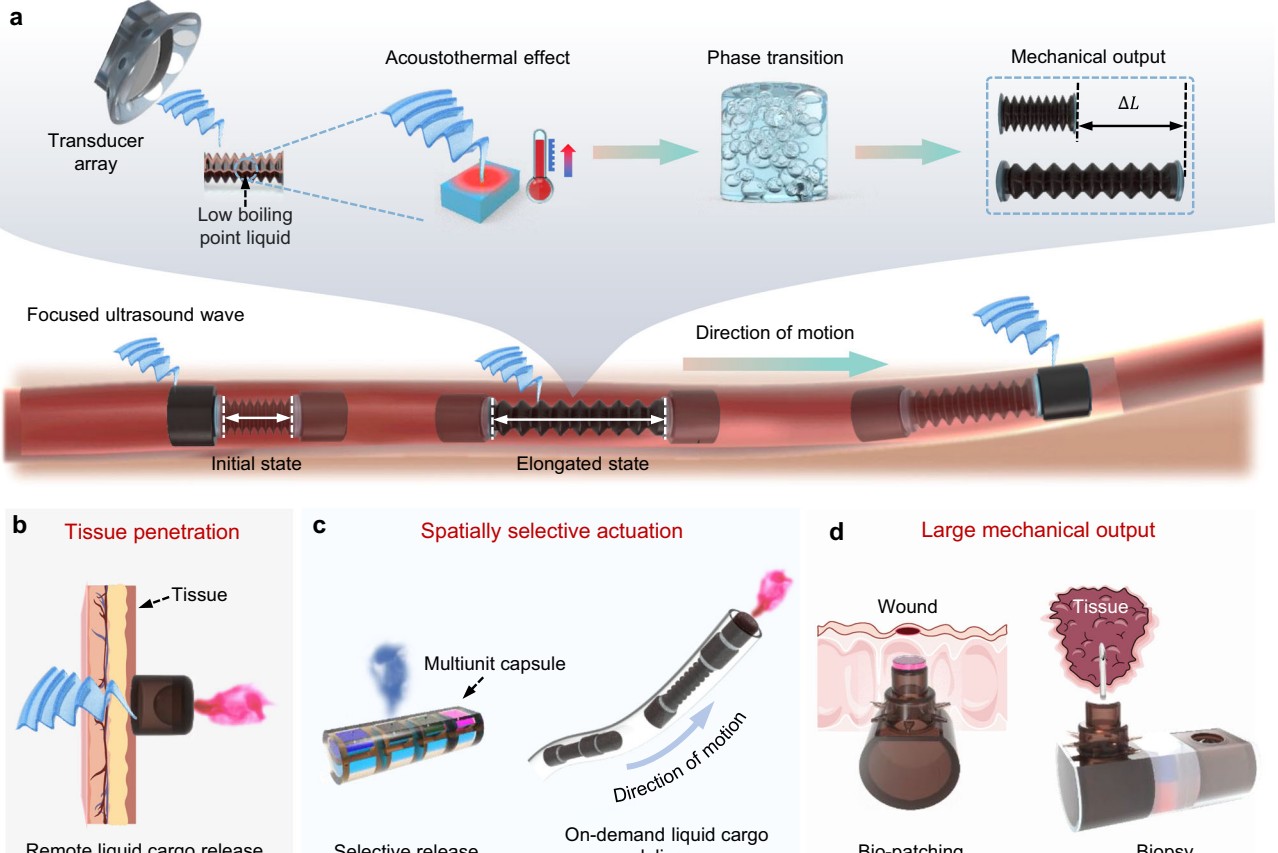

**Fig. 1 | Mechanism of focused ultrasound-controlled phase transition (FUPT) actuation method. a** Schematic illustration of the FUPT actuation method. **b** Tissue penetration ability of the FUPT method. The soft capsule can be triggered by the focused ultrasound field through tissue. **c** Spatially selective actuation ability of the FUPT method. The multiunit capsule and in-pipe soft robot can achieve selective and on-demand liquid cargo delivery, respectively. **d** Large mechanical output ability of the FUPT method. The soft robot can be actuated to execute tissue patching and biopsy tasks inside the intestine.

The acoustothermal effect induced phase transition and structural design endow the deformation and motion ability of actuators/robots. To achieve superior energy conversion efficiency, the acoustic impedance of the actuator needs to match with the surrounding medium. The acoustic impedance ($Z$) can be calculated as follows:

$$Z = \rho S \quad (1)$$

where $\rho$ and $S$ are the density and sound speed of the medium, respectively. Considering a normal incident ultrasound wave crosses an interface between two materials with different impedances, the reflected and transmitted energy can be calculated as follows:

$$R = \left(\frac{z_2 - z_1}{z_2 + z_1}\right)^2 \quad (2)$$

$$T = \frac{4z_2 z_1}{(z_2 + z_1)^2} \quad (3)$$

where $R$ and $T$ are the reflection and transmission coefficients ($R + T = 1$), respectively. $z_1$ and $z_2$ are the acoustic impedance of two materials. According to Eq. (3), more wave energy will be transmitted than reflected when the acoustic impedances of the two materials are matched ($z_1 \approx z_2$). In our study, the transmission coefficient between $Fe_3O_4NPs$ doped Ecoflex 00-30 with water (similar to most biofluid environments) and tissue are 98.7% and 96.8%, respectively (detailed

calculations can be found in Supplementary Section 2), indicating that the acoustic transparency of the proposed material in underwater or biological environments and its application potentials.

To investigate the influence of particle doping on the acoustothermal effect, we measured the temperature change of elastomer samples with and without doping of $Fe_3O_4NPs$ under the same acoustic field. The experimental setup can be found in Supplementary Section 3. As illustrated in Fig. 2d, the composite material experiences rapid heating due to the acoustothermal effect, this property can be utilized to induce the phase transition of enclosed low boiling point liquids. In addition, the comparison of the temperature change curves reveals that the heating rate of the $Fe_3O_4NPs$-doped Ecoflex (~14.9 °C/s, estimated by the slope of the tangent line at the initial position) is faster than that of pure Ecoflex (~8.0 °C/s), indicating that the introduction of $Fe_3O_4NPs$ can accelerate the response of the soft actuators. This can be attributed to two potential reasons: first, the doping of $Fe_3O_4NPs$ with high thermal conductivity may improve the heat transfer rate (5.5 W m$^{-1}$K$^{-1}$ for $Fe_3O_4NPs$ with 20 nm diameter[41] compared to 0.16 W m$^{-1}$K$^{-1}$ for pure Ecoflex 00-30[42]), which will result in faster heat conduction through the materials. Second, the presence of $Fe_3O_4NPs$ can enhance the acoustic attenuation of the material[43,44] (the attenuation coefficient of pure Ecoflex and $Fe_3O_4NPs$ doped Ecoflex at 1.7 MHz are 0.54 dB mm$^{-1}$ and 0.84 dB mm$^{-1}$, respectively, detailed calculation can be found in Supplementary Section 4), resulting in higher acoustic-to-thermal conversion efficiency. For the same reason, doping $Fe_3O_4NPs$ can achieve a higher steady-state temperature under the same ultrasound excitation (Fig. 2e). Furthermore, the higher

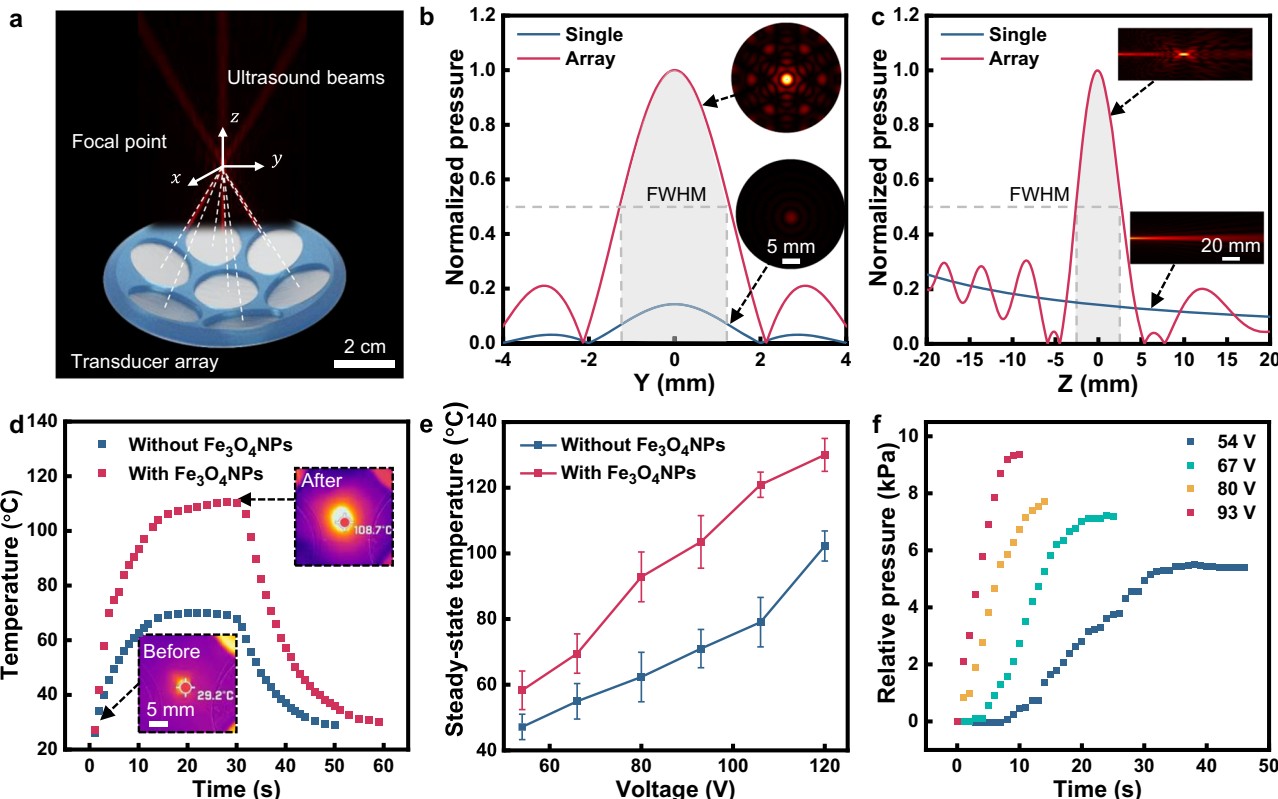

**Fig. 2 | Characterization and influence factors of FUPT actuation method. a** PZT transducer array and generated acoustic field. **b** Comparison of the simulated lateral acoustic pressure distribution and (**c**) longitudinal acoustic pressure distribution between a single transducer and a transducer array. The shaded area indicates the full-width half maximum (FWHM) amplitude region. **d** Temperature change of polymer films during the ultrasound activation period with and without Fe$_3$O$_4$NPs doping. **e** Steady-state temperature for polymer film with and without Fe$_3$O$_4$NPs doping. Error bars represent the standard deviation ($n = 3$). **f** Relative internal pressure change of FUPT-based actuator under different peak-to-peak actuation voltages.

energy input (i.e., higher excitation voltage) can produce a higher steady-state temperature (Fig. 2e) and pressure change inside the actuator caused by phase transition (Fig. 2f), thus enabling bigger deformation and force output of soft actuators.

## Structural design and applications of FUPT-based actuators

Based on the acoustothermal effect and the high spatial resolution of FUPT, a series of soft actuators and multifunctional soft robots with selective actuation ability are demonstrated. As shown in Fig. 3a–d, FUPT actuated expansion and elongation actuators are designed and validated using both finite element simulations and experimental verification. For the expansion actuator, a pillar is incorporated to connect the upper and lower surfaces at the center (Fig. 3a, Supplementary Fig. 11a, and Supplementary Movie 1). This structural design facilitates lateral expansion while minimizing elongation. It is important to note that the maximum Von Mises stress (33544 N/m$^2$) represents the tension within the soft actuator, and the contact pressure between the actuator and the surrounding medium (e.g., soft tissue) is usually much lower than this value. If the cross-sectional area is smaller than the area of the surrounding environment, the contact pressure can be zero even the actuator is fully expanded. Similarly, the longitudinal elongation with minimal expansion is achieved through a varying cross-section structure design (Fig. 3c, Supplementary Fig. 11b, and Supplementary Movie 1). This design allows for larger elongation while limiting the degree of lateral expansion. The actuation voltage plays a crucial role in controlling the rate of expansion (diameter change) and elongation (length change) of the actuators, higher actuation voltage results in faster deformation of the actuators (Fig. 3b, d). The nature of soft pneumatic actuators enables significant deformation, making them versatile foundational components for soft robots with enhanced functionalities.

By integrating the above-mentioned actuators with expansion and elongation deformations, we have demonstrated the in-pipe untethered soft robot with selective actuation ability for pipeline exploration in the liquid environment. Which finds wide usage in industrial[45] and medical fields. For instance, implanted catheters (with diameters ranging from several millimeters[46] to a dozen millimeters[47]) play a crucial role in drainage, and in-pipe robots may help with regular maintenance to reduce the risk of blockage or infection[48,49]. Specifically, this robot consists of two expansion actuators for anchoring and one elongation actuator for length adjustment (Fig. 3e and Supplementary Movie 1), these actuators can be independently controlled by mechanically moving the focal point of the ultrasound field. The locomotion can be divided into several steps as follows: firstly, expansion actuator A is activated to anchor the lower terminal of the robot to the pipe ((I)–(II) shown in Fig. 3e–g). Secondly, the elongation actuator B is excited to stretch the robot body ((II)–(III) shown in Fig. 3e–g). Subsequently, the expansion actuator C is activated to anchor the upper end of the robot to the pipe ((III)–(IV) shown in Fig. 3e–g). This process continues until actuators A and B are cooled, completing one cycle of locomotion and allowing the soft robot to reach a new location, as shown in (IV)–(V) in Fig. 3e–g. Each cycle of actuation corresponds to one stride motion, and the repeatability of the phase transition process enables the FUPT-based soft robot to achieve long-distance locomotion. The moving direction control of the robot can be realized by changing the activation sequence of the consisted actuators, as illustrated in Fig. 3h and Supplementary Movie 1, the fabricated robot first overcomes gravity to move upward and followed by downward motion in the pipe. Figure 3i

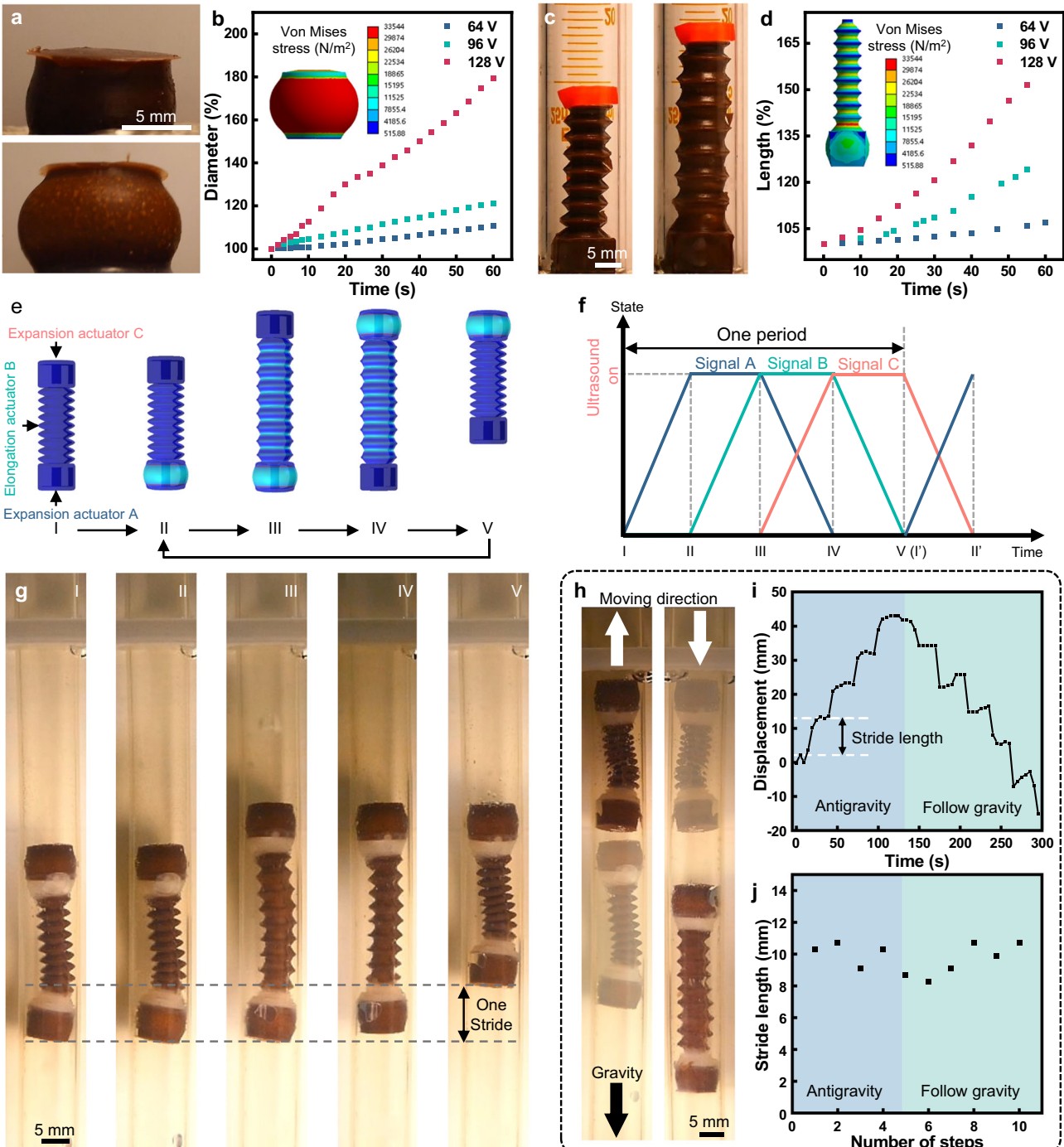

**Fig. 3 | FUPT-based actuators and in-pipe untethered soft robot. a** Photographs of the FUPT-based expansion actuator. **b** Diameter change of the expansion actuator with different actuation voltage. Inset: simulated deformation of the expansion actuator. **c** Photographs of the FUPT-based elongation actuator. **d** Length variation of the elongation actuator with different actuation voltages.

Inset: simulated deformation of the elongation actuator. **e** Schematic diagram and corresponding (**f**) applied signals and (**g**) images showing the operating principle and motion of the in-pipe robot. **h** Overlaid images showing the upward and downward movement of the in-pipe soft robot. **i** Displacement and (**j**) stride length for every cycle of the in-pipe robot obtained from Supplementary Movie 1.

shows the displacement of the robot during the upward and downward motion process obtained from Supplementary Movie 1, indicating the stability of its locomotion and the anchoring performance of the expansion actuators. It is worth noting that even in soft biological pipes with the presence of mucus, the robot still possesses anchoring ability, as shown in Supplementary Fig. 12. However, there is still room for improvement in enhancing the robot's movement robustness, enabling it to effectively address the challenges posed by the soft and folded nature of biological pipelines and further ensure the possibility

of successful operation of the robots in diverse and more challenging environments. Additionally, the negligible deviation in displacement for each cycle, as demonstrated in Fig. 3j, attests to the excellent actuation repeatability of the FUPT-based soft robot. This repeatability is crucial for consistent and reliable performance in various applications.

The combination of the tissue penetration capability offered by the FUPT method with the functional actuator design holds potential for ultrasound-actuated robots in biomedical applications. Figure 4a

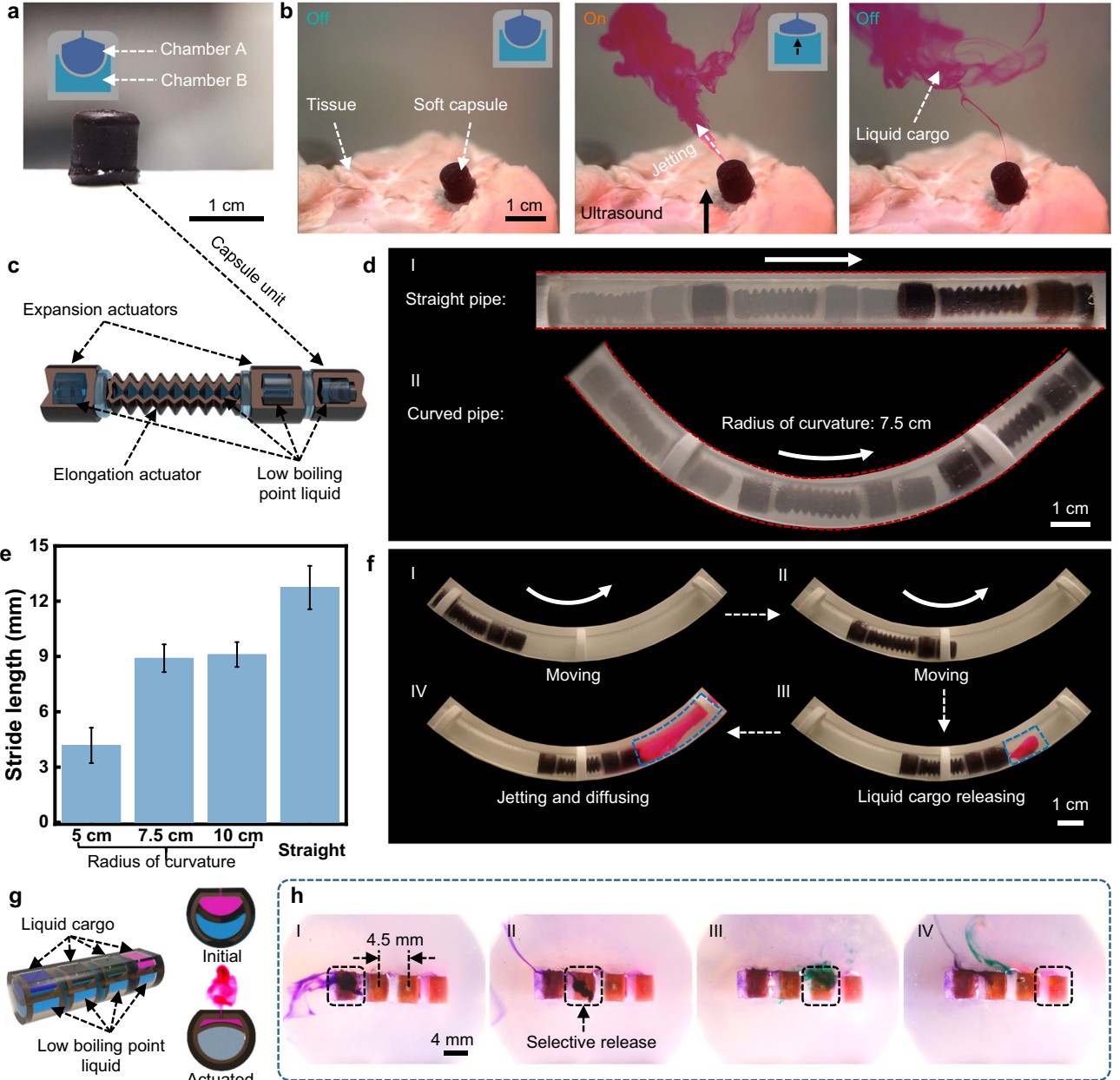

**Fig. 4 | FUPT-based on-demand liquid cargo delivery soft robot and multiunit soft capsule. a** Structural design and image of the capsule unit. **b** A series of images showing the ultrasound actuated liquid cargo release through the tissue (ultrasound waves originate from the bottom and travel through the tissues to the soft capsule). **c** Schematic image showing the structure of the in-pipe soft robot with a capsule unit. **d** Overlaid images of the robot that moving in the straight and curved pipes. **e** Relationship between the stride length of the robot and the radius of curvature of the pipes. Error bars represent the standard deviation ($n = 3$). **f** Overlaid images showing the FUPT-driven on-demand liquid cargo delivery and release process. **g** Structural design of the multiunit soft capsule. **h** A series of images demonstrating the selective liquid cargo release by aligning the high-intensity acoustic field with different chambers.

shows a liquid cargo (e.g., drug) storage and release system in the form of a capsule unit, the capsule consists of two individual chambers, where chamber A is filled with liquid cargo and chamber B is filled with low boiling point liquid. As illustrated in Fig. 4b and Supplementary Movie 2, ultrasound can penetrate through tissues to remotely activate chamber B and expel the loaded liquid.

To demonstrate the multifunctionality and application potential of the FUPT method in soft robots, we integrate the capsule unit with the pipeline robot (Fig. 4c). As shown in Fig. 4d and Supplementary Movie 2, the capsule-carrying robot can navigate through straight or curved pipes. The stride length of the robot depends on the curvature radius of the pipes, with a larger radius resulting in longer stride lengths (Fig. 4e). In addition, the

capsule unit can be individually triggered and realize the powerful jetting of the loaded cargo (Fig. 4f).

Furthermore, the high resolution of the FUPT method proves beneficial when selectively activating a specific unit among multiple similar units is required. We have fabricated the multiunit soft capsule with each unit measuring approximately 4 mm × 4 mm × 3.5 mm, as depicted in Fig. 4g and Supplementary Movie 2. By aligning the focal point of the acoustic field with different units, selective release of one specific liquid cargo can be achieved (Fig. 4h). The experimental results demonstrate that FUPT can achieve selective actuation of adjacent soft actuators with a central distance of approximately 4.5 mm. Which might be useful for applications where multiple medications are required[22].

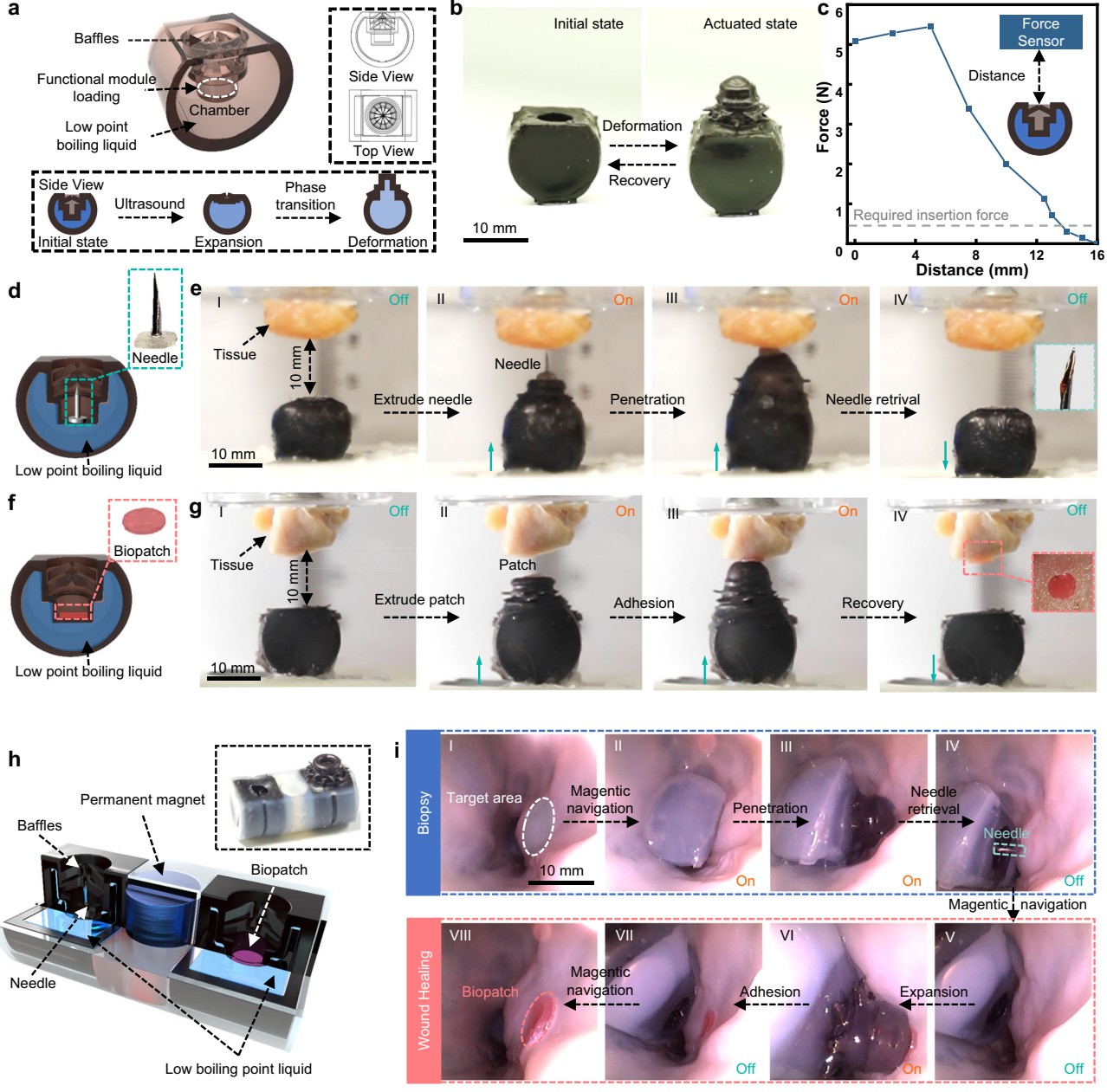

**Fig. 5 | FUPT-powered soft robot for biopsy and tissue patching. a** Schematic illustration of the structure of the soft actuator for biopsy and tissue patching. **b** Photographs of the soft actuator before and after actuation. **c** Relationship between actuation force and the initial distance between the actuator and the target plane. **d** Schematic illustration of the biopsy soft actuator. **e** A series of images illustrating the biopsy procedure. **f** Schematic illustration of the tissue patching soft actuator. **g** A series of images demonstrating the tissue patching procedure. **h** Schematic illustration of the structural design and the image of the soft robot for biopsy and tissue patching. **i** A series of endoscopic images demonstrating the biopsy and tissue patching procedure of the soft robot inside the intestine.

Another essential advantage of the FUPT actuation method is its ability to deliver large mechanical output in terms of deformation and output force. For instance, Fig. 5a demonstrates a soft device designed for in-body biopsy and patching. In its initial state, the closed baffles can prevent the functional tools (e.g., needle and bio-patch) from contact with the external environment (Supplementary Fig. 13). This is crucial for biopsy and patching procedures as leakage or contamination of bio-samples can lead to complications. Upon actuation, the functional tool inside the actuator is extruded and interacts with the target tissues. When the internal pressure decreases, the actuator returns to its initial state, retracting the functional tool back into the soft actuator (Fig. 5b). We also measured the maximum output force of the soft actuator in different initial places, as shown in Fig. 5c. The

detected force increases as the distance increase from 0 mm to 5 mm, potential reason is that before reaching 5 mm, the internal folding structure of the actuator may not have fully unfolded, resulting in residual force within the system that has not been exerted. The biggest force output is approximately 5.5 N at a distance of 5 mm. This output force meets the requirement for most minimally invasive surgeries[21]. Continuously increasing the distance between the actuator and the target leads to a decrease in the detected force because when the actuator is fully actuated, the top is truncated cone-shaped and the diameter of the cross-section decreases as it approaches the top end of the actuator. Consequently, the output force provided under the same pressure is lower as the contact area between the actuator and the target is reduced. Note that even at a distance exceeding 13 mm

(considering the length of the needle, this actuator can perform tissue sampling approximately 20 mm away from the target), the actuator can still provide enough force for the needle to be inserted in the tissue (detailed measurements can be found in Supplementary Section 5). This indicates that the FUPT method allows for utilizing smaller sized actuators for sampling based on the requirements of different application environments (i.e., varying luminal diameters), thus reducing patient discomfort during the task performing.

The significant mechanical output ability of the FUPT-actuated soft actuators forms a foundation for many biomedical applications. As shown in Fig. 5d, f, a needle, and a bio-patch are loaded in the soft actuators as functional tools for biopsy and tissue patching, respectively. The soft actuator undergoes substantial deformation when it is remotely activated by ultrasound with less than 10 s ((I)–(II) in Fig. 5e, g), causing the internal functional tools to be extruded towards the target. Continuous ultrasound stimulation allows for the insertion of the needle and adhesion of bio-patch ((III) in Fig. 5e, g). Finally, after being deactivated the ultrasound field for 220 s, the actuator returns to its initial state ((IV) in Fig. 5e, g), completing the tissue sampling and patching tasks (Supplementary Movie 3 and Supplementary Section 5). It is worth noting that the cooling time is influenced by the temperature difference between the actuator and the environment, the thermal conductivity of the materials used, and the structural geometry (the detailed discussion can be found in Supplementary Section 6).

Considering the real application scenarios of the FUPT-based soft actuator, we have evaluated its stability and temperature sensitivity in simulated human temperature environment. As shown in Supplementary Fig. 14, the actuator can maintain a stable state in human body temperature condition, while also achieving actuation when stimulated by focused ultrasound (as shown in Supplementary Fig. 15). Notably, instead of undergoing sudden volume expansion at 34 °C (i.e., the boiling point of the Novec 7000 at 1 atm), the phase transition of the internal material is a gradual process. With rising temperature, the saturation vapor pressure of the material increases, leading to a higher proportion of liquid vaporization. In other words, as the temperature rises, the phase transition leads to an increase in pressure and a higher boiling point for the remaining liquid, which ensures the stability of the soft robot in human body's temperature environment. Furthermore, the FUPT-based actuator exhibits dynamic sensitivity to the environmental temperature changes. The higher temperature has been found to induce greater sensitivity (Supplementary Fig. 14c). Within the range of 25 °C to 37 °C, the actuator experiences an approximate 50% increase in internal volume. At around 42.5 °C, successful actuation is achieved as the internal part was extruded by the actuator. As the temperature continued to rise to 45 °C (the temperature that normal cells can withstand[50]), the internal volume expanded by 220%. Additionally, the utilization of focused ultrasound provides an additional advantage in terms of thermal safety. Thanks to the high spatial resolution of the focused ultrasound field, the focal point of the ultrasound can be positioned inside the soft actuator, which induces the limited surface temperature (i.e., contact with tissues. As shown in Supplementary Fig. 16) increase when compared with the higher temperature inside the actuator's body in actuated state and will not cause damage to surrounding tissues. The dynamic response ability to temperature changes and high spatial resolution of the focused ultrasound makes the FUPT-based soft actuator suitable for biomedical applications.

Additionally, a multi-functional soft robot can be realized by combining soft actuators with different functions and an axially magnetized cylindrical permanent magnet. As shown in Fig. 5h, a permanent magnet is placed between the sampling actuator and tissue patching actuator, allowing for adjustable orientation angles of the robot as required (Supplementary Fig. 17). Combined with the biomedical imaging method, the soft robot can employ magnetic navigation and ultrasound actuation to accomplish precise target biopsy

and in-situ tissue patching in the intestine (Fig. 5i and Supplementary Movie 3). The demonstrated application scenarios suggest the necessity and effectiveness of the proposed FUPT actuation method, which benefits from the combination of tissue penetration ability, selective actuation ability, and large mechanical output.

## Imaging and control systems for the FUPT actuation method

The precise selective actuation for soft robots applied in biomedical fields puts forward higher requirements for spatial localization methods, especially in the confined space inside the human body where optical imaging is inaccessible for providing visual feedback. Therefore, we demonstrated a control system by combining the FUPT actuation method and ultrasound imaging[51–53]. As shown in Fig. 6a, b, a focused ultrasound transducer array is integrated with an ultrasound imaging probe, both mounted at the end of a robotic arm. Therefore, the proposed control system consists of three subsystems (Fig. 6c, d): the motion system (robotic arm), imaging system (ultrasound probe), and actuation system (focused ultrasound transducer array). The workflow of this control system is illustrated in Fig. 6c, and Supplementary Movie 4 as follows: first, the robotic arm moves along the scanning path while the ultrasound probe captures images along the path. These images are then input into the YOLOv8 network for instance segmentation. Once the soft robot is detected, its relative coordinates with respect to the ultrasound probe can be calculated, allowing for the determination of the central position of the soft robot. Moreover, the 3D reconstruction and registration of the soft robot provides its orientation information, which can be utilized to adjust the direction of needle insertion and allows for calculating the optimal position and orientation of the imaging plane to sense the state of the robot. Based on the position and orientation information, the robotic arm automatically adjusts its position to align the focal point of the acoustic field with the soft robot. Finally, the FUPT-based soft robot can be actuated using the proposed system (Fig. 6eI–II). The actuation state of the robot can be monitored by ultrasound imaging, which enables on-off control over the duration of the acoustic field action (Fig. 6eII–IV), reducing unnecessary acoustic radiation exposure. It is important to note that in real-world applications, the acoustic properties of the surrounding medium may differ from pure water. When there is a need to stimulate the actuator deep inside the body, the penetration depth of ultrasound can be ameliorated by increasing the number of the transducers in the array. This increase not only enhances the power output, but also improves the numerical aperture (NA). In our experiments, we successfully achieve the actuation of the soft robot through tissues with a thickness of approximately 8 cm, as demonstrated in Supplementary Fig. 18, which demonstrates the remarkable tissue penetration ability and application potentials of the proposed FUPT method. Additionally, the presence of inhomogeneous medium with different acoustic impedance (e.g., the air within body cavities) between the ultrasound source and robot poses challenges for precise actuation. Current solution, such as ultrasound contrast agents[54,55], can be employed to create windows for ultrasound waves and effectively resolves the issue.

We also demonstrate the biopsy tasks in the intestine using the proposed imaging and control system (Fig. 6f, g). In this experiment, the ultrasound passes through the tissue and stimulates the soft robot inside the intestine, allowing for needle insertion and recovery. The integration of ultrasound imaging (or other biomedical imaging method as shown in Supplementary Fig. 19) and real-time monitoring enables accurate positioning and control of the soft robot in the tissue environment, ensuring efficient and safe operation within biological tissues.

## Discussion

In this study, we have introduced a focused ultrasound-controlled phase transition (FUPT) method for the actuation of untethered soft

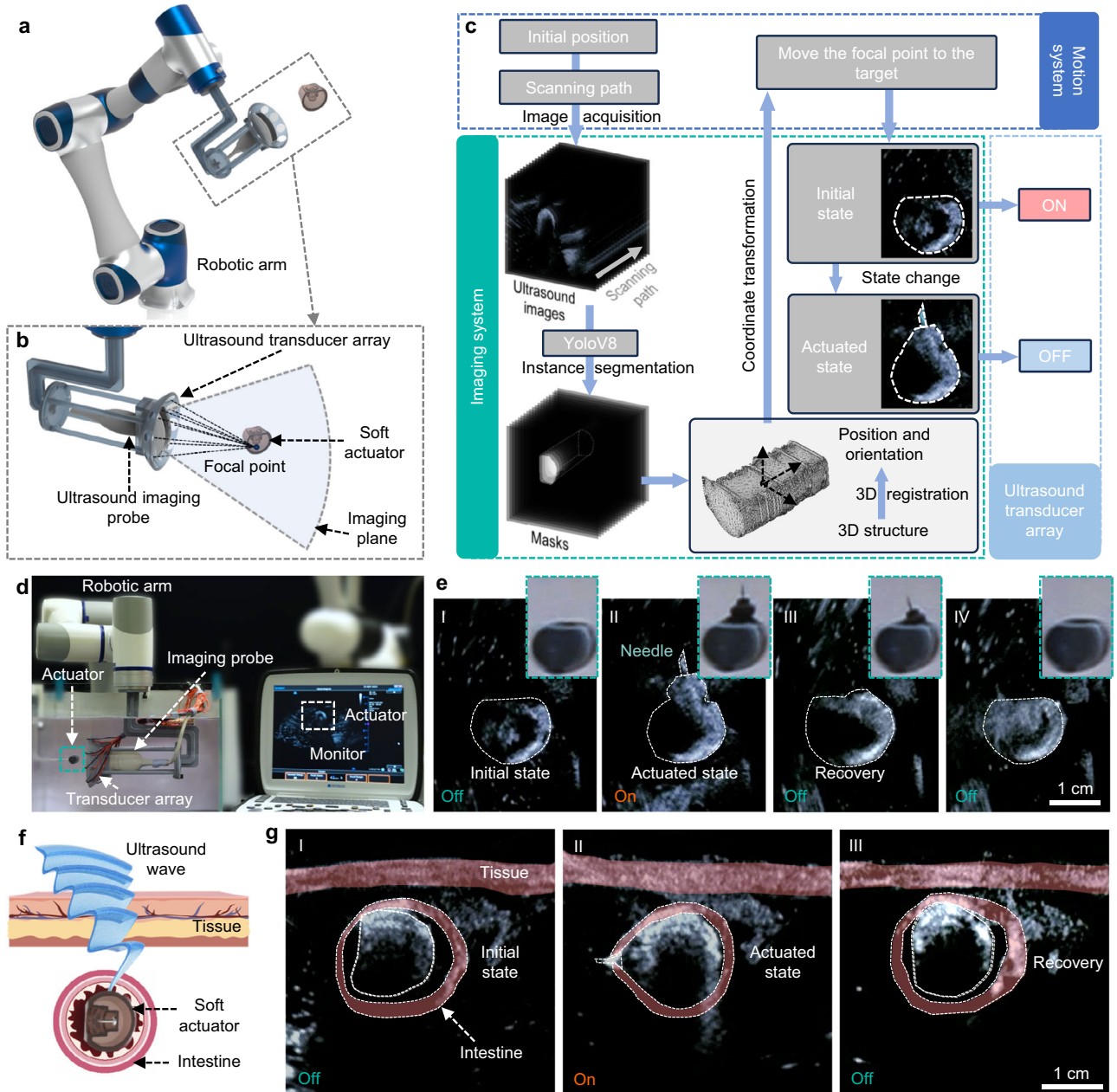

**Fig. 6 | Imaging and control system for FUPT actuated soft robot. a, b** Diagram and (**c**) workflow of the proposed control system. **d** Photograph of the experimental setup. **e** A series of ultrasound and optical images of the soft actuator during actuation in water environment. **f** Schematic and (**g**) ultrasound images illustrating the biopsy process controlled by the proposed system.

robots. This method harnesses the unique advantages of ultrasound in terms of its ability to penetrate biofluids and tissues while enabling precise selective actuation with millimeter-level resolution. The FUPT actuation method also can provide substantial mechanical output, enabling the FUPT-based actuator to generate Newton-level forces. These characteristics make the FUPT method well-suited for diverse applications, especially those in the biomedical and underwater environments.

The effectiveness of the FUPT actuation method is demonstrated through various actuators capable of simple deformation patterns like expansion and elongation. By combining these actuators, we have designed an untethered in-pipe robot with locomotion ability that realized the on-demand liquid cargo delivery and controllable release. Additionally, we have designed a soft robot specifically for biopsy and tissue patching applications, emphasizing the versatility and robust

mechanical output of our proposed method. To achieve precise control, we have developed a control system incorporating ultrasound imaging for automatically aligning the acoustic field focal point with the soft robot, and implemented on-off control based on the observation of the robot's state. It is worth noting that the proposed FUPT actuation method can seamlessly integrate with magnetic resonance thermometry, enabling precise and near real-time temperature control of the robot. This integration significantly expands the potential applications of FUPT-based medical soft robotics, opening up new possibilities for advanced medical procedures.

While the FUPT method is demonstrated to be a noteworthy advance for the actuation of untethered soft robot, to ensure the adaptability of the FUPT-based soft robot for scenarios that involve a large number of actuation cycles or long-term implanted devices, we also propose further exploration to for lowering the vapor

permeability. This could involve investigating alternative matrix that exhibit improved vapor sealing properties, as discussed in Supplementary Section 7. Furthermore, future research can focus on enhancing the environmental adaptability of FUPT-driven soft robots through customized designs tailored to specific application scenarios. For instance, when designing robots for operation in high-speed fluid-filled blood vessels, it is essential to consider the mechanical and thermal effects of the fluid on the robot. These explorations would broaden the diversity and functionality of FUPT-based soft robots. We envision that the FUPT strategy could provide a basis for the actuation of untethered miniature soft robots that can further enlarge their applications in the biomedical and industrial fields.

## Methods
### Acoustic field analysis
The acoustic field was generated by arranging the array of lead zirconate titanate (PZT) piezoelectric transducers oriented towards a calculated focal point[56]. Each transducer (piezoelectric ceramic plates with 20 mm diameter and 1.2 mm thickness, Alibaba Co., Ltd, China) operated in its first-order longitudinal vibration mode with a resonance frequency of 1.7 MHz. The piston source model was introduced to simulate the wave emitted by each transducer, the complex acoustic pressure $P$ at position vector $\mathbf{r}$ (i.e., the vector from the centre of the emitting surface of the piston source to the calculated position in space) is governed by[57,58]:

$$P(\mathbf{r}) = P_0 V \frac{D_f(\theta)}{d} e^{i(kd + \phi_0)} \tag{4}$$

where $P_0$ is an amplitude constant that specifies the transducer output power (details of the acoustic pressure amplitude calibration can be found in Supplementary Section 1), $V$ is the applied voltage to the transducer, $d = \|\mathbf{r}\|$ is the propagation distance in free space, $\phi_0$ is the initial phase of the transducer, the wavenumber $k$ is determined by $k = 2\pi/\lambda$, where $\lambda$ is the wavelength. The directivity function of the piston source $D_f(\theta)$ is calculated through $D_f(\theta) = 2J_1(ka\sin\theta)/ka\sin\theta$, where $J_1$ is a first-order Bessel function of the first kind, $a$ is the radius of the piston. In addition, the total acoustic field is derived by summing the contributions of each transducer in the arrays:

$$P_{total}(\mathbf{r}) = \sum_{i=1}^{n} P(\mathbf{r}) \tag{5}$$

### Fabrication of the soft actuators
The phase transition soft actuator was fabricated by template method with 3D printed molds (LD-002, Shenzhen Creality 3D Technology Co., Ltd.). The actuator has a hollow structure composed of $Fe_3O_4$NPs doped elastomer (Ecoflex 00-30, Smooth-On Inc.) as the shell and is filled with low boiling point liquid, which is the pressure origination based on the liquid-gas phase transition process. Note that additional surface treatment was indispensable for the printed molds so that the silicone polymerization process could be entirely complete[59]. Details can be found in Supplementary Section 8.

### Finite element analysis
Finite element analysis (FEA) simulations were conducted using Ansys software (Ansys Inc., Canonsburg, PA, USA) to predict and optimize the deformation of the soft actuators. Static structural analysis was employed, and the material properties of the polymer used can be found in Supplementary Section 9. During the simulations, the inflation process was simulated by applying normal pressure to all internal surfaces of the actuators. This approach allows for the evaluation of the deformation and response of the actuators under the applied pressure.

### Imaging and control system
The imaging and control system consists of a robotic arm (Dobot CR5, China), an ultrasound imaging system (Terason t3200, Teratech Corporation, USA) with a curved array imaging probe (5C2A, Teratech Corporation, USA), and a homemade transducer array. The ultrasound image is transmitted to the PC via an image acquisition card. Python is used for coding, while socket is employed to communicate with the robotic arm.

### Three-dimensional reconstruction and orientation determination of soft robots
We trained a YOLOv8 network for instance segmentation of ultrasound images frame by frame, extracting the cross-sectional mask of the robot. Subsequently, each mask is discretized into scattered points. After stacking the scattered points layer by layer, the three-dimensional point cloud of the soft robot is obtained. Next, to determine the orientation information of the soft robot, the iterative closet point (ICP) algorithm is employed for the registration of the real 3D structure and the reconstructed structure (Supplementary Fig. 20). Typically, the ICP algorithm iterates between two steps:
1. Identification of the correspondence set $\mathcal{K} = \{(\mathbf{p},\mathbf{q})\}$ from target point cloud $\mathbf{P}$, and source point cloud $\mathbf{Q}$, which has been transformed with current transformation matrix $\mathbf{T}$.
2. Update the transformation $\mathbf{T}$ by minimizing an objective function $E(\mathbf{T})$ defined over the correspondence set $\mathcal{K}$.

In our situation, we initially calculated the centroids of both the real 3D structure (source point cloud $\mathbf{Q}$) and the reconstructed 3D structure (target point cloud $\mathbf{P}$). These two centroids were overlapped to set the initial position for the ICP registration. For faster convergence speed, we leveraged the point-to-plane ICP algorithm, using the following objective function[60]:

$$E(\mathbf{T}) = \sum_{(\mathbf{p},\mathbf{q}) \in \mathcal{K}} \left( (\mathbf{p} - \mathbf{T}\mathbf{q}) \cdot \mathbf{n_p} \right)^2 \tag{6}$$

where $\mathbf{n_p}$ is the normal of point $\mathbf{p}$. After completing the registration, the orientation information of the 3D reconstructed structure is obtained, which enables us to determine the optimal ultrasound imaging plane for observing the actuation state of the soft robots.

### Coordinate transformation
To calculate the position and posture of the robotic arm based on ultrasound imaging, coordinate transformation was performed. Supplementary Fig. 21 illustrates the reference frames involved in this process. The world frame for the robot base is denoted as {W}, the frame for the robot flange is {F}, and the ultrasound imaging plane frame is represented as {U}. By utilizing these frames, a coordinate transformation matrix from the ultrasound imaging plane frame to the world frame can be obtained.

$$\mathbf{T}_W^U = \mathbf{T}_W^F \mathbf{T}_F^U \tag{7}$$

relative position $\mathbf{P}_a^U \in \mathbb{R}^3$ in the ultrasound frame to the world frame $\mathbf{P}_a^W = [x_a^W, y_a^W, z_a^W] \in \mathbb{R}^3$ can be derived as:

$$[\mathbf{P}_a^W] = \mathbf{T}_W^U[\mathbf{P}_a^U; 1] \tag{8}$$

### Reporting summary
Further information on research design is available in the Nature Portfolio Reporting Summary linked to this article.

## Data availability
All data generated in this study are provided in the article and its Supplementary Information.

## Code availability
All the relevant code in this paper is available upon request from the corresponding author.

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

## Acknowledgements

The research work is financially supported by the Hong Kong Research Grants Council (RGC) with project Nos. R4015-21, RFS2122-4S03, the RGC Collaborative Research Fund (CRF) with Project No. C1134-20GF; the Croucher Foundation Grant with Ref. No. CAS20403, and the CUHK internal grants. The authors also thank the support from the Multi-scale Medical Robotics Centre (MRC), InnoHK, at the Hong Kong Science Park, and the SIAT-CUHK Joint Laboratory of Robotics and Intelligent Systems.

## Author contributions

B.H. and L.Z. conceived and initiated the project. B.H. and X.W. carried out the experiments. B.H. and M.S. conducted the analysis. B.H. and X.W. analysed the data. B.H., X.W., Y.C., C.Z. and L.S. carried out the fabrication. B.H. wrote the code. B.H., X.W. and Y.D. wrote the manuscript. C.X., H.Y., J.Z. and X.L. assisted in research discussion and article writing. L.Z., Y.D. and B.L. provided supervision from technical and writing aspects and supported the research. B.H., X.W. and Y.D. contributed equally to this work.

## Competing interests

The authors declare no competing interests.
