## [Peer Review File · Nature Communications]

REVIEWER COMMENTS

Reviewer #1 (Remarks to the Author):

Hao et al. present the engineering of a series of soft robots actuated by ultrasound, which takes advantage of focused ultrasound's ability to penetrate tissue. This is an innovative concept with interesting clinical applications. The manuscript is written in a clear fashion with supplementary videos to demonstrate the workings of the device.

To help envision the next step of translating this technology into clinical studies, I hope the authors can address these questions:

- What is the sensitivity of the robot to environmental/temperature changes during its real-life application
- How will movement of the device be manipulated in order to ensure precise area/direction is biopsied or patched
- what is the range of penetration depth of ultrasound at which the robot can be controlled
- What is the timeframe of the recovery phase and in which the robot could be positioned, as certain tasks can be time sensitive, e.g. retracting the needle, and retrieval of the robot containing biopsied tissue
- can ultrasound be applied precisely in real-life scenario to actuate the robot with the presence of inhomogeneous medium between the probe and robot, especially air in the bowel

Reviewer #2 (Remarks to the Author):

This paper presents a focused ultrasound-controlled phase transition (FUPT) strategy for achieving millimetre-level spatially selective actuation and Newton-level force of soft robots operating within tissue or liquid environments. A soft robot for on-demand liquid cargo delivery in the pipe and a biopsy robot for tissue acquisition and patching in the intestine are designed to verify their application potential. A control system combining the FUPT actuation method and ultrasound imaging is demonstrated to achieve precise operations. These works are interesting and the results have application potential in many fields. However, several issues need to be addressed:

1. As we all know, the body temperature is between 36 °C and 37.7 °C. Will the low boiling liquid used in this article automatically evaporate in the human body?
2. In Fig. 3b, the maximum mises stress of the mechanism can reach 33544 N/m². Some internal organs of the human body may not be able to withstand such high pressure, such as blood vessels. Authors may consider designing improvement methods for the above phenomena or providing special explanations for their applied range.
3. Does the presence of mucus in actual biological pipelines affect the anchor of expansion actuators, especially in the case of anti-gravity?
4. Will high Reynolds number environments (such as fast flow rates) have a significant impact on robot action execution?
5. Is it possible to make a table to compare the results of this study with other research results and point out the advantages of this study?

Reviewer #3 (Remarks to the Author):

This paper reports a series of soft robots that can be actuated wirelessly and selectively using focused ultrasound. The most amazing part of this work is that the soft robots can generate Newton-level force output. The authors achieve this by using acoustothermal effect to heat a low-boiling-point liquid above its phase transition temperature using ultrasound. In this way, the soft robots proposed by the authors can behave like other pneumatic or hydraulic soft robots, but in a wireless way without the need of tubes. The authors doped silicone elastomer with Fe₃O₄ nanoparticles to enhance the acoustothermal effect, and demonstrated many applications such as on-demand liquid cargo delivery, biopsy, and tissue patching. Overall, the mechanism is novel, the soft robots show clear advantages in output force, and the demonstrations are sufficient to prove the conclusions claimed by the authors. Therefore, I recommend publication of this work in Nature Communications. However, I have the following questions.

My biggest concern is the envisioned biomedical applications. Since the core body temperature varies by about 1°C each day, the liquid used for the robot needs to have a boiling point a few degrees higher than the body temperature. Therefore, to actuate the robot in vivo, there will be a local temperature increase of a few °C. Considering that the robot is in millimeter to centimeter scale, will the temperature increase in such a large area cause some side effects inside the body? It's fine to use the low-boiling-point liquid with a boiling point of 34°C for demonstration, but how will the authors deal with the local temperature increase in vivo?

A related question is how accurate can the authors control the temperature on the soft robots? Since the ultrasound wave can be partially reflected by bone or other intermediate medium, the energy delivered to the robots may not be the same in different scenarios. How to deal with this issue, especially for biomedical applications where slight temperature increase may have significant effect on the tissues.

In addition, soft materials are usually permeable to gas or even liquid. How to avoid the leakage of gas during heating? Will this cause safety issue for in vivo applications? How about the actuation performance of the soft robots in multiple cycles?

Some minor questions.

- Can the authors compare the force (or force per gram/volume) and response time (or cooling time) of their robots with other approaches, such as embedding a shape memory alloy in soft elastomers? It's better to add a figure or table to compare the performances.
- The reviewer wonders about the biocompatibility of the Novec 7000 engineering fluid.
- Can the authors be more quantitative on selective actuation? Since there is thermal diffusion, the two actuated parts cannot be too close. What is the minimal distance between different actuation parts to guarantee selective actuation?

Response to referees

Reviewer #1

Hao et al. present the engineering of a series of soft robots actuated by ultrasound, which takes advantage of focused ultrasound's ability to penetrate tissue. This is an innovative concept with interesting clinical applications. The manuscript is written in a clear fashion with supplementary videos to demonstrate the workings of the device.

Thank you for your careful review and valuable feedback of this work. We appreciate your positive remarks regarding the innovation and application potential of our focused ultrasound-actuated soft robots. We have addressed each comment point-by-point and made the appropriate changes in the revised version of manuscript.

To help envision the next step of translating this technology into clinical studies, I hope the authors can address these questions:

1. What is the sensitivity of the robot to environmental/temperature changes during its real-life application.

Response:

The authors appreciate the reviewer's comments. As the reviewer's suggestion, we tested the stability of the actuator in human body's temperature and the temperature sensitivity of the FUPT-based soft actuator. Considering its biomedical application scenarios that should avoid the potential damage, the temperature range were set from 25 °C to 45 °C. Note that the human body's temperature is approximately 37 °C (within the test range) and the 45 °C is the highest temperature that most normal cells can withstand¹. To simulate environmental temperature variations, we employed a water bath heating method, as shown in Fig. R1.1 (Supplementary Fig. 14). The actuator was sealed inside a container, and the temperature was accurately monitored using a thermometer. The volume changes of the actuator were measured through the water level change in a glass tube connected to the top of the container (Fig. R1.1a). To

validate the stability of the soft actuator in a human body temperature environment, we conducted experiments as shown in Fig. R1.1b, where the internal volume of the actuator remained unchanged after being exposed to around 37 °C for over 10 minutes, indicating that the FUPT based robotics can keep a steady state. Furthermore, the temperature sensitivity of the actuator is demonstrated in Fig. R1.1c. As the temperature increased from 25 °C to 37 °C, there was an approximate 50% increase in the internal volume of the actuator. At around 42.5 °C, the internal part of the actuator can be extruded, representing successful actuation. As the temperature continued to rise to 45 °C, the internal volume expanded by 220%. These results indicate that the internal volume of the soft actuator continues to increase as the temperature rises, accompanied by an increasing inflation rate that correlated with the temperature (represented by an increasing slope), which indicates that the FUPT-based actuator has the dynamic sensitivity that responds to the environmental temperature changes, the higher the temperature induced the higher sensitivity. At 25 °C, the volume change rate is about 2.6% per degree, whereas at 45 °C, it increases to about 36.7% per degree (as calculated from Fig. R1.1c). Note that the phase transition of the enclosed fluid is a gradual process which originates from the increase in the saturation vapor pressure with increasing temperature. This effect ensures that there will be no sudden volume expansion at 34 °C (i.e. boiling point of Novec 7000 at 1 atm), and make our designed robot can be utilized in the environments of human body's temperature, which was further verified by extruding and retrieving the internal part of the soft actuator that controlled by the ultrasound stimulation in 37.5 °C environment, as shown in Fig. R.1.2 (Supplementary Fig. 15).

Fig. R1.1 (Supplementary Fig. 14): Temperature sensitivity measurement of the soft actuator. (a) Photograph of the experiment setup and (b) stability of the soft actuator in 37 °C environment. (c) The

internal volume change of the soft actuator as the temperature increased.

Fig. R1.2 (Supplementary Fig. 15): The images of the actuator extruding and retrieving the function tool actuated by ultrasound in 37.5 °C environment.

We have added the above results as **Supplementary Fig. 14 and 15**, and added the corresponding discussion on **Page 18, line 304** as follows:

“Considering the real application scenarios of the FUPT-based soft actuator, we have evaluated its stability and temperature sensitivity in simulated human temperature environment. As shown in Supplementary Fig. 14, the actuator can maintain a stable state in human body temperature condition, while also achieving actuation when stimulated by focused ultrasound (as shown in Supplementary Fig. 15). Notably, instead of undergoing sudden volume expansion at 34 °C (i.e. the boiling point of the Novec 7000 at 1 atm), the phase transition of the internal material is a gradual process. With rising temperature, the saturation vapor pressure of the material increases, leading to a higher proportion of liquid vaporization. In other words, as the temperature rises, the phase transition leads to an increase in pressure and a higher boiling point for the remaining liquid, which ensures the stability of the soft robot in human body’s temperature environment. Furthermore, the FUPT-based actuator exhibits dynamic sensitivity to the environmental temperature changes. The higher temperature has been found to induce greater sensitivity (Supplementary Fig. 14c). Within the range of 25 °C to 37 °C, the actuator experiences an approximate 50% increase in internal volume. At around 42.5 °C, successful actuation is

achieved as the internal part was extruded by the actuator. As the temperature continued to rise to 45 °C (the temperature that normal cells can withstand⁵⁰), the internal volume expanded by 220%.”

2. How will movement of the device be manipulated in order to ensure precise area/direction is biopsied or patched.

Response:

We thank the reviewer for making this point. In order to realize precise biopsy and patching in the desired area and direction, we have implemented magnetic manipulation to accurately control the movement and rotation of the soft robot. By utilizing magnetic gradient force, we can control the locomotion of the robot, allowing it to navigate to the target area. Additionally, magnetic moment is employed to manipulate the rolling motion and to control the orientation of the robot, as shown in Fig. R1.3 (Supplementary Fig. 17). This allows for the accurate alignment of the robot with the target area for biopsy or patching, as shown in Supplementary Video 3.

Fig. R1.3 (Supplementary Fig. 17): (a) Schematic illustration showing the orientation adjustment of the soft robot controlled by external magnetic field. (b) The soft robot aligns its direction with the orientation of the external magnetic field at (b) 0°, (c) 180°, (d) -45°, (e) -90° and (f) -135°, respectively.

In addition, imaging technology is essential to provide feedback for precise manipulation. Information of the robot's spatial position and orientation plays a vital role in achieving accurate biopsy or tissue patching. In our study, we conducted experimental verification using endoscope, ultrasound imaging, 3D reconstruction, and 3D registration methods to acquire the soft robot's spatial information, as depicted in Fig. 5, Fig. 6, Supplementary Fig. 20, and Supplementary Video 4.

We have revised the manuscript on **Page 19, line 329** as follows:

“Combined with the biomedical imaging method, the soft robot can employ magnetic navigation and ultrasound actuation to accomplish precise target biopsy and in-situ tissue patching in the intestine (Fig. 5i and Supplementary Video 3). The demonstrated application scenarios suggest the necessity and effectiveness of the proposed FUPT actuation method, which benefits from the combination of tissue penetration ability, selective actuation ability, and large mechanical output.”

3. What is the range of penetration depth of ultrasound at which the robot can be controlled.

Response:

We thank the reviewer's comment regarding the penetration depth of ultrasound waves. The penetration depth of ultrasound can be ameliorated by increasing the number of the transducers in the array, the increase of the transducers not only enhances the power output but also improves the numerical aperture (NA). In our former experiments (Fig. 4b in the manuscript), we successfully excited the soft actuator through a tissue with approximately 4 cm thickness (i.e. the penetration depth ~ 4 cm). To further showcase the adjustability of the focused ultrasound device, we have fabricated a new ultrasound transducer array with an increased penetration depth of ~ 8 cm. This was achieved by incorporating 36 transducers in one array with a focal length of 10 cm, as illustrated in Fig. R1.4 (Supplementary Fig. 18).

Fig. R1.4 (Supplementary Fig. 18): Ultrasound penetration experiment. (a) Top view and (b) side view of a focused ultrasound transducer array with 36 elements. (c) Schematic illustration the experimental setup for assessing ultrasound penetration depth. (d) Initial state and (e) actuated state of the soft actuator before and after ultrasound excitation through tissue with the thickness of ~ 8 cm.

We conducted experiments to assess the tissue penetration ability of the ultrasound waves using the larger transducer array. Chicken tissue was placed between the transducers and the soft actuator, as depicted in Fig. R1.4c - e. The results revealed that when the tissue thickness reached to ~ 8 cm, the soft actuator could still be excited by ultrasound waves, as shown in Fig. R1.4d and R1.4e. It is worth mentioning that existing FDA-approved HIFU devices² have better capability to treat tumors located within 10 cm from the skin surface³. Considering the thickness of the human body, this range covers a significant volume ratio of the body. Thus, we believe that the FUPT method can directly benefit from existing medical devices, leading to improved accessibility and performance.

We have added the above results as **Supplementary Fig. 18**, and added the corresponding discussion on **Page 21, line 360** as follows:

“It is important to note that in real-world applications, the acoustic properties of the surrounding medium may differ from pure water. When there is a need to stimulate the actuator deep inside

the body, the penetration depth of ultrasound can be ameliorated by increasing the number of the transducers in the array. This increase not only enhances the power output, but also improves the numerical aperture (NA). In our experiments, we successfully achieve the actuation of the soft robot through tissues with a thickness of approximately 8 cm, as demonstrated in Supplementary Fig. 18, which underscores the remarkable tissue penetration ability and application potentials of the proposed FUPT method.”

4. What is the timeframe of the recovery phase and in which the robot could be positioned, as certain tasks can be time sensitive, e.g. retracting the needle, and retrieval of the robot containing biopsied tissue.

Response:

The authors appreciate the reviewer’s comment. The positioning time of the soft robot demonstrated in Fig. 5i can be long enough because the location of the soft robot are controlled by stable and large magnetic gradient force, which results in sufficiently long positioning time. In addition, as the reviewer’s suggestions, we have conducted the recovery time measurement of the actuator as shown in Fig. R1.5 (Supplementary Fig. 6). The duration required for the biopsy and patching tasks of the fabricated soft robot that demonstrated in Fig. 5 ranges from 80 s to 220 s (Fig. R1.5). It should be noted that the recovery time can be reduced, and the influence factors were specifically discussed as follows:

The recovery speed is influenced by various factors, such as the temperature difference between the actuator and the environment, the thermal conductivity of the materials involved, and the geometry of the structure. For example, the actuator’s recovery time in air is approximately 220 s at 25 °C, as shown in Fig. R1.5a. However, when the actuator is submerged in water, the time can be shortened to 80 s, as water ($0.58 \text{ Wm}^{-1}\text{K}^{-1}$) has much higher thermal conductivity compared to air ($0.024 \text{ Wm}^{-1}\text{K}^{-1}$). Furthermore, the larger the temperature difference between the environment and the soft robot, the faster the thermal conduction and the shorter cooling

time. For instance, the soft actuator can be cooled down to an unactuated state within 80 s when immersed in 25 °C water (Fig. R1.5b), while the time is extended to 110 s in 37.5 °C (Fig. R1.5c).

Fig. R1.5 (Supplementary Fig. 6): Comparison of recovery time (a) in air, 25 °C, (b) in water, 25 °C, and (c) in water 37.5 °C.

In addition, the larger thermal conductivity of the embedded materials (e.g. $\text{Fe}_3\text{O}_4\text{NPs}$) enabled the soft robot to exhibit higher thermal conduction capability thus inducing the faster cooling, as demonstrated in Fig. 2d. Furthermore, the heating and cooling time is also influenced by the size of the actuator, which follows the scaling law. The required amount of heat is proportional to the volume (which scales with the cube of the characteristic size), while the heat dissipation area corresponds to the square of the characteristic size. As a result, smaller actuators exhibit faster heating and cooling rates, as the heating and cooling speed is inversely proportional to the size. For instance, as shown in Fig. 4g and 4h, the 4 mm-sized actuator achieves actuation and recovery within a few seconds, which is noticeably faster than the larger actuator depicted in Fig. 4a. These findings highlight the scale-related nature of these processes and provide potential solutions to enhance the response speed of the soft robot.

We have edited the manuscript and added discussion on **Page 17, line 298** in the manuscript as follows:

“Finally, after being deactivated the ultrasound field for 220 s, the actuator returns to its initial state ((IV) in Fig. 5e and 5g), completing the tissue sampling and patching tasks (Supplementary Video 3 and Supplementary Section 5). It is worth noting that the cooling time is influenced by the temperature difference between the actuator and the environment, the thermal conductivity of the materials used, and the structural geometry (the detailed discussion can be found in Supplementary Section 6).”

And we also added the above results on **Supplementary Section 6** and the corresponding discussion for the influence factors of cooling time shown as follows:

“Supplementary Section 6. Influence factors of cooling time for the soft actuator

The timeframe of the recovery phase is a crucial aspect to consider when evaluating the applicability of the presented soft robot for time-sensitive applications.....

To summarize, the recovery phase of the soft robot can be adjusted by considering several factors. Immersion in a liquid environment, such as water, accelerates the cooling process, while the choice of materials, thickness of the structure, and size of the actuator also influence the heating and cooling rates. By optimizing these parameters, the soft robot can be tailored to meet the requirements of time-sensitive applications.” (Details can be found in Supplementary section 6 in the revised Supplementary information)

5. Can ultrasound be applied precisely in real-life scenario to actuate the robot with the presence of inhomogeneous medium between the probe and robot, especially air in the bowel.

Response:

The authors appreciate the reviewer's comments. As the reviewer's concern, ultrasound encounters challenges when it comes to penetrating through air due to the impedance mismatch at the gas-water interface. Similarly, the presence of bones causes refraction of sound waves, making it difficult to produce a focused ultrasound field. Nevertheless, these challenges have been properly addressed and have not hindered the widespread of ultrasound in medical settings.

First, in cases involving bones, leveraging the acoustic holographic lens⁴ or precisely controlling the phase of the transducer array can correct aberrations caused by the presence of an inhomogeneous medium, such as the skull bone⁵. These phase adjustment methods can successfully address the distortion issues arising from medium heterogeneity and improve focusing ability. Second, in the presence of gas in lumen or cavity inside the body, a common method in the real medical treatment is to fill with liquid to create an acoustic window. For example, during abdominal ultrasound examination in the hospital, patients are instructed to drink 500 ml to 1000 ml of water to fill the bladder, thus creating an acoustic window for ultrasound to pass through. In addition, through drinking contrast medium with high viscosity, the air in the gastrointestinal tract can be displaced and providing an acoustic window that lasts about 60 minutes⁶, thus allows ultrasound to penetrate the stomach⁶ or intestines⁷. The water/contrast medium drinking method to exhaust air can be easily applied to realize the precise control and imaging for our FUPT-based robot.

We envision that these existing techniques, along with ongoing advancements, can well combined with proposed FUPT actuation method in this work and further enhance the application potentials and clinical translation for untethered medical soft robots.

In the manuscript, we address the reviewer's suggestions by adding the discussion on **Page 21, line 366** as follows:

“Additionally, the presence of inhomogeneous medium with different acoustic impedance (e.g. the air within body cavities) between the ultrasound source and robot poses challenges for precise actuation. Current clinical medical treatment, such as ultrasound contrast agents,^{54,55} can be employed to create windows for ultrasound waves and effectively resolves the issue.”

Reviewer #2

This paper presents a focused ultrasound-controlled phase transition (FUPT) strategy for achieving millimeter-level spatially selective actuation and Newton-level force of soft robots operating within tissue or liquid environments. A soft robot for on-demand liquid cargo delivery in the pipe and a biopsy robot for tissue acquisition and patching in the intestine are designed to verify their application potential. A control system combining the FUPT actuation method and ultrasound imaging is demonstrated to achieve precise operations. These works are interesting and the results have application potential in many fields. However, several issues need to be addressed:

We thank the reviewer for careful review and high evaluation of our work. Your constructive comments have helped us to further improve our manuscript. We have addressed every comment and made corresponding changes to the manuscript. Please check the following point-by-point response for details.

1. As we all know, the body temperature is between 36 °C and 37.7 °C. Will the low boiling liquid used in this article automatically evaporate in the human body?

Response:

We thank the reviewer for making this point. In fact, the soft actuator can operate normally when working in environments close to human body temperature. To provide evidence of its operability in such temperature condition, we conducted a series of experiments to assess the stability and performance of the soft actuator in environments with human body temperature.

To simulate real-world application scenarios, we employed a water bath heating method and sealed the actuator inside a container. The temperature inside the container was accurately monitored using a thermometer, and the volume changes of the actuator were measured through the water level change in a glass tube connected to the top of the container (Fig. R2.1a,

Supplementary Fig. 14). During the experiment, we observed that the internal volume of the actuator remained unchanged when exposed to a temperature close to 37 °C for over 10 minutes, as shown in Fig. R2.1b. The reason behind this behavior lies in the fact that, under such temperature conditions, a small portion of the low-boiling point liquid within the actuator undergoes a phase change, causing an increase in internal pressure. As a result, the boiling point of the remaining liquid increases, effectively halting vaporization and reaching a dynamic equilibrium within the actuator. This characteristic ensures the stability and functionality of the actuator even in environments exceeding 37 °C. In addition, the actuator can still be actuated by ultrasound to perform its intended functions in human body temperature conditions, as demonstrated in Fig. R2.2 (Supplementary Fig. 15).

Fig. R2.1 (Supplementary Fig. 14): Temperature sensitivity measurement of the soft actuator. (a) Photograph of the experiment setup and (b) stability of the soft actuator in 37 °C environment.

Fig. R2.2 (Supplementary Fig. 15): The images of the actuator extruding and retrieving the function tool actuated by ultrasound in 37.5 °C environment.

We have added the above results as **Supplementary Fig. 14-15**, and added the corresponding discussion on **Page 18, line 304** as follows:

“Considering the real application scenarios of the FUPT-based soft actuator, we have evaluated its stability and temperature sensitivity in simulated human temperature environment. As shown in Supplementary Fig. 14, the actuator can maintain a stable state in human body temperature condition, while also achieving actuation when stimulated by focused ultrasound (as shown in Supplementary Fig. 15). Notably, instead of undergoing sudden volume expansion at 34 °C (i.e. the boiling point of the Novec 7000 at 1 atm), the phase transition of the internal material is a gradual process. With rising temperature, the saturation vapor pressure of the material increases, leading to a higher proportion of liquid vaporization. In other words, as the temperature rises, the phase transition leads to an increase in pressure and a higher boiling point for the remaining liquid, which ensures the stability of the soft robot in human body’s temperature environment.”

2. In Fig. 3b, the maximum mises stress of the mechanism can reach 33544 N/m². Some internal organs of the human body may not be able to withstand such high pressure, such as blood vessels. Authors may consider designing improvement methods for the above phenomena or providing special explanations for their applied range.

Response:

The authors appreciate the reviewer’s comment. We clarify that the Von Mises stress depicted in the Fig. 3b and 3d represents the maximum tension within the soft material of the actuator, which is much higher than the contact pressure with tissues. As shown in Fig. R2.3, the value of maximum Von mises stress (maximum tension in actuator’s skin) is higher than the internal pressure, and the internal pressure is still higher than the contact pressure between the actuator and the surrounding environment (e.g. organs and tissue). For example, in a balloon catheter used intravascularly, the contact pressure applied to the blood vessel is less than 10% of the balloon’s internal pressure, and the increase in contact pressure (i.e. the pressure applied to the

vessel) is less than 1% of the increased balloon pressure⁸. Additionally, if the balloon is not in contact with the blood vessel, the contact pressure acting on the vessel is zero no matter how big the balloon pressure is⁸. Therefore, by considering the type of tissue being contacted and utilizing feedback from ultrasound visualization (as illustrated in Fig. 6e and 6g), controlling the maximum volume expansion of the actuator can effectively ensure the safety of its contact with tissues.

Fig. R2.3: Schematic of the force relationship of internal pressure, tension in the actuator’s skin (Von miss stress) and contact pressure.

To improve the clarity, we have revised the manuscript on **Page 11, line 185** shown as follows:”

“It is important to note that the maximum Von Mises stress (33544 N/m^2) represents the tension within the soft actuator, and the contact pressure between the actuator and the surrounding medium (e.g. soft tissue) is usually much lower than this value. If the cross-sectional area is smaller than the area of the surrounding environment, the contact pressure can be zero even the actuator is fully expanded.”

3. Does the presence of mucus in actual biological pipelines affect the anchor of expansion actuators, especially in the case of anti-gravity?

Response:

We appreciate the reviewer's comment. As the reviewer's suggestions, we conducted experiments to investigate the anchor performance of the expansion actuator in the presence of mucus in a biological pipeline (i.e. esophagus).

As shown in Fig. R2.4 (Supplementary Fig. 12), the pipeline robot (about 3 g) was placed inside an esophagus which was vertically suspended, the blocks (each block weights 6.2 g) were fixed at the robot's body as loading to test the anchor ability in the esophagus with mucus. In the unexpanded state without the ultrasound stimulation, the pipeline robot lacked the ability to anchor within the esophagus and would slide down within 1 s due to the reduced friction force caused by mucus (the loading is ~ 4 times of the weight of robot) (Fig. R2.4a). As shown in Fig. R2.4b, once the actuator was expanded, the robot was able to maintain relative stability with twice the load for over 15 s (i.e. four blocks of 24.8g that 8.3 times than the self-weight of the robot), indicating the effectiveness of the expandable actuator in the actual biological pipelines with the presence of mucus.

Fig. R2.4 (Supplementary Fig. 12): Anchoring capacity of the pipeline robot in the esophagus with loadings in (a) unexpanded and (b) expanded state.

However, we recognize that there are additional challenges to navigating in such complex environments. Natural biological pipelines contain folds and obstacles that poses requirement of robustness in the robot's movements. For instance, the elongation actuator should provide

significantly greater output force and stiffness to overcome the hindrance caused by the folds and ensure smooth forward motion.

We have added the above results as **Supplementary Fig. 12** and corresponding discussion on **Page 13, line 224** as follows:

“It is worth noting that even in soft biological pipes with the presence of mucus, the robot still possesses anchoring ability, as shown in Supplementary Fig. 12. However, there is still room for improvement in enhancing the robot's movement robustness, enabling it to effectively navigate the challenges posed by the soft and folded nature of biological pipelines and further ensure the possibility of successful operation of the robots in diverse and more challenging environments.”

4. Will high Reynolds number environments (such as fast flow rates) have a significant impact on robot action execution?

Response:

The authors appreciate the reviewer's comment. It is true that the motion of the pipeline robot demonstrated in Fig. 3 and 4, can be affected by high Reynolds number environments. This is because the motion of the robot relies on the anchoring of the expansion modules to the pipe, which can obstruct fluid flow when in the expanded condition. On the other hand, for the biopsy and patching robot shown in Fig. 5 and 6, functions of biopsy and tissue patching are achieved through the internal pressure change induced by the phase transition of the liquid inside the robot body. In such cases, the high Reynolds number environments have limited influence on these functions. However, there is no denying that the magnetic driven locomotion of the robot will be influenced if the fluidic forces are high enough. In the current study, we are primarily focused on the FUPT actuation method, and the demonstrated robots are specifically designed to the scenario with limited fluid flow, such as the gastrointestinal tract.

In addition, we totally agree with the reviewer's concern regarding the potential application of the FUPT method in environments characterized by high Reynolds number, such as blood vessels. To achieve such goal, future work should meticulously consider the robot's structural design, material properties, and combination with other control strategies. For instance, one potential amelioration is to design the anchor actuator in such a way that it expands in a specific direction or is hollow when in an activated state, instead of completely occupying the entire pipe. This design modification would create space for fluid to pass through even when the actuator is expanded.

We have added a brief discussion on **Page 23, line 400** for future exploration related with concern, shown as follows:

“Furthermore, future research can focus on enhancing the environmental adaptability of FUPT-driven soft robots through customized designs tailored to specific application scenarios. For instance, when designing robots for operation in high-speed fluid-filled blood vessels, it is essential to consider the mechanical and thermal effects of the fluid on the robot. These explorations would broaden the diversity and functionality of FUPT-based soft robots.”

5. Is it possible to make a table to compare the results of this study with other research results and point out the advantages of this study?

Response:

The authors appreciate the reviewer's comment and totally agree with the suggestion of a comparison table can further strengthen and highlight the advantages of this work. We have compared the proposed FUPT method with other actuation strategy in different aspects as shown in Table R2.1 (Supplementary Table 3).

We believe that for the development of the untethered soft robotic system, the energy input and the energy conversion processes are the most important aspects, the former determines the total

amount of energy that can be supplied to the robot in a given environment, while the latter corresponds to the conversion efficiency of the energy.

First, from the perspective of the energy input process, the untethered actuation methods based on “waves” (or dynamic fields) can benefit from energy accumulation and enabling heating capabilities. Although the response time may be longer compared to quasi-static fields, it becomes easier to achieve better output performance. This is evident in the force comparison between magnetic soft robots actuated by quasi-static fields and the FUPT actuation method, as shown in Table R2.1. In addition, the proposed FUPT method in this work is based on the mechanical waves, which offers superior spatial resolution while ensuring tissue penetration depth compared to electromagnetic (EM) waves. This is evident in the spatial selective actuation ability and tissue penetration depth comparison between EM waves and the FUPT actuation method, as shown in Table R2.1.

Secondly, regarding the energy conversion process, the liquid-gas phase transition method has been successfully employed in previous studies, with reported output forces reach up to 70 N⁹. Embracing this effective principle, the proposed FUPT actuation method exhibited the excellent force output (i.e. Newton’s level) satisfying the output force requirements in many medical scenarios, such as biopsy, stent deployment, or suturing.

Therefore, the proposed FUPT actuation method realized the combination of the high spatial resolution, high tissue penetration depth and the enough force output, which enables the FUPT-based untethered soft robot with great potentials in biomedical application and clinical translation.

We have included the comparison table into the Supplementary Information shown as follows:

Table R2.1 (Supplementary Table 3): The comparison of various actuation methods for untethered soft actuator and robot.

Strategy		Selective actuation	Newton-level force	Response time	Scale	Centimeter-level tissue penetration	Medical image	Ref
Quasistatic field	Magnetic soft robot	No	No (~ 60 μ N) ¹⁹	Much less than one second	Millimeter	Yes	Yes (Ultrasound)	8
		No		Much less than one second	Millimeter to centimeter	Yes	No	9
Electromagnetic wave	Liquid crystal elastomer	Yes (Spatial)	No	Less than one second	Submillimeter	No (~ 0.03-0.05 cm) ^{10,11}	No	12
		Yes (frequency)	No	A few seconds to tens of seconds	Millimeter to centimeter	No (~ 0.05-0.3 cm) ^{10,11}	No	13
	Laser-induced phase transition	Yes (Spatial)	N/A	Tens of seconds	Centimeter	No (~ 0.5 cm) ^{10,11}	No	14
	Millimeter wave-induced phase transition	Yes (Spatial)	Yes (38 N)	Ten seconds to tens of seconds	Centimeter	No (~ 0.05 cm) ¹⁵	No	16
	Shape memory alloy	Yes (frequency)	No (0.6 N)	A few seconds	Millimeter to centimeter	Yes	No	17
	Shape Memory Polymer	No	No	Ten seconds to tens of seconds	Millimeter to centimeter	Yes	No	18
	Coiled artificial muscle	No	Yes (~3.1 N)	A few seconds	Millimeter to centimeter	Yes	No	19
	Radio frequency-induced phase transition	No	Yes (3.1 N to 70 N)	A few seconds	Millimeter	Yes	Yes (Ultrasound)	20
No		Yes (up to 31 N)	Ten seconds to tens of seconds	Centimeter	Yes	No	21	
Mechanical wave	Focused ultrasound-controlled phase transition	Yes (Spatial)	Yes (up to 5.5 N)	A few seconds to tens of seconds	Millimeters to centimeter	Yes (~ 8 cm)	Yes (Ultrasound, X-ray)	This work

Reviewer #3 (Remarks to the Author):

This paper reports a series of soft robots that can be actuated wirelessly and selectively using focused ultrasound. The most amazing part of this work is that the soft robots can generate Newton-level force output. The authors achieve this by using acoustothermal effect to heat a low-boiling-point liquid above its phase transition temperature using ultrasound. In this way, the soft robots proposed by the authors can behave like other pneumatic or hydraulic soft robots, but in a wireless way without the need of tubes. The authors doped silicone elastomer with Fe₃O₄ nanoparticles to enhance the acoustothermal effect, and demonstrated many applications such as on-demand liquid cargo delivery, biopsy, and tissue patching. Overall, the mechanism is novel, the soft robots show clear advantages in output force, and the demonstrations are sufficient to prove the conclusions claimed by the authors. Therefore, I recommend publication of this work in Nature Communications. However, I have the following questions.

We thank the reviewer for careful review and high evaluation of this work. Your constructive comments and suggestions have helped us to further improve our manuscript. We have addressed each comment point-by-point and made the appropriate changes in the revised version of manuscript.

1. My biggest concern is the envisioned biomedical applications. Since the core body temperature varies by about 1°C each day, the liquid used for the robot needs to have a boiling point a few degrees higher than the body temperature. Therefore, to actuate the robot in vivo, there will be a local temperature increase of a few °C. Considering that the robot is in millimeter to centimeter scale, will the temperature increase in such a large area cause some side effects inside the body? It's fine to use the low-boiling-point liquid with a boiling point of 34 °C for demonstration, but how will the authors deal with the local temperature increase in vivo?

Response:

We thank the reviewer for making this point. We have conducted the experiments shown as Fig. R3.1 and R3.3 (Supplementary Fig. 14-16) to address this concern. The results demonstrate the following findings: (1) the FUPT based soft devices can maintain a steady state and operate normally with the temperature of application scenario (i.e. the human's body temperature). (2) Through the use of focused ultrasound with high spatial resolution, we have found that the temperature increase is primarily localized within the body of the actuator. As a result, the surface temperature exhibits limited rising compared to the temperature increase inside the actuator during the actuation process controlled by ultrasound. This temperature rise will not cause damage to the surrounding tissues.

We first analyzed the phase transition behavior of the enclosed low boiling point liquid in human body's temperature conditions. To simulate the temperature of the real application scenario, we employed a water bath heating method and sealed the actuator inside a container, the temperature inside the container was accurately monitored using a thermometer. The volume changes of the actuator were measured through the water level change in a glass tube connected to the top of the container, as shown in Fig. R3.1a (Supplementary Fig. 14a). The internal volume of the actuator remained unchanged during being exposed to a temperature close to 37 °C for over 10 minutes (as shown in Fig. R3.1b), indicating the evaporation stopped inside the robot. This could be attributed to the phase change process of the Novec 7000 engineering fluid. Initially, a small portion of the low boiling liquid undergoes phase transition within the robot body, resulting in an increase in internal pressure and subsequently raising the boiling point of the internal liquid. This, in turn, leads to a higher response temperature. Furthermore, the soft actuator demonstrated the ability to be actuated by ultrasound and perform its functions effectively at an environmental temperature of 37.5 °C, as demonstrated in Fig. R3.2 (Supplementary Fig. 15).

Moreover, as shown in Fig. R3.1c, we demonstrated the temperature sensitivity of the actuator. As the temperature increased from 25 °C to 37 °C, there was an approximate 50% increase in the internal volume of the actuator. Successful actuation occurred when the temperature

reached around 42.5 °C, as the internal part was extruded by the actuator. Furthermore, as the temperature continued to rise to 45 °C, the internal volume expanded by 220%.

Fig. R3.1 (Supplementary Fig. 14): Temperature sensitivity measurement of the soft actuator. (a) Photograph of the experiment setup and (b) stability of the soft actuator in 37 °C environment. (c) The internal volume change of the soft actuator as the temperature increased.

Fig. R3.2 (Supplementary Fig. 15): The images of the actuator extruding and retrieving the function tool actuated by ultrasound in 37.5 °C environment.

In addition, we have tested the temperature changes of the tissue and robot's surface when exposed to focused ultrasound. As shown in Fig. R3.3 (Supplementary Fig. 16), we employed an ultrasound transducer array to generate localized heating inside the actuator through the tissue, and temperature changes were measured using a thermometer. When ultrasound waves were focused on the actuator for about 60 s, the temperature probe placed between the tissue and the actuator recorded a temperature of approximately 39.1 °C (starting from 37 °C). It is important to note that at this point, the soft actuator had already been actuated, indicating that its internal temperature (i.e. >42.5 °C according to the temperature sensitivity tests results

shown in Fig. R3.1c) was considerably higher than the surface temperature (i.e. 39.1 °C). This phenomenon can be attributed to the high spatial resolution of the focused ultrasound, ensuring that the focal point of the ultrasound was situated inside the soft actuator. Therefore, the limited temperature change of the FUPT-based robot's surface temperature will not cause damage on the contacted tissue.

Fig. R3.3 (Supplementary Fig. 16): Temperature changes in tissue (in contact with the actuator) when the actuator is actuated by ultrasound waves through tissue.

We have added the above results as **Supplementary Fig. 14-16**, and added the corresponding discussion on **Page 18, line 304** as follows:

“Considering the real application scenarios of the FUPT-based soft actuator, we have evaluated its stability and temperature sensitivity in simulated human temperature environment. As shown in Supplementary Fig. 14, the actuator can maintain a stable state in human body temperature condition, while also achieving actuation when stimulated by focused ultrasound (as shown in Supplementary Fig. 15). Notably, instead of undergoing sudden volume expansion at 34 °C (i.e. the boiling point of the Novec 7000 at 1 atm), the phase transition of the internal material is a gradual process. With rising temperature, the saturation vapor pressure of the material increases, leading to a higher proportion of liquid vaporization. In other words, as the temperature rises, the phase transition leads to an increase in pressure and a higher boiling point for the remaining liquid,

which ensures the stability of the soft robot in human body's temperature environment. Furthermore, the FUPT-based actuator exhibits dynamic sensitivity to the environmental temperature changes. The higher temperature has been found to induce greater sensitivity (Supplementary Fig. 14c). Within the range of 25 °C to 37 °C, the actuator experiences an approximate 50% increase in internal volume. At around 42.5 °C, successful actuation is achieved as the internal part was extruded by the actuator. As the temperature continued to rise to 45 °C (the temperature that normal cells can withstand⁵⁰), the internal volume expanded by 220%. Additionally, the utilization of focused ultrasound provides an additional advantage in terms of thermal safety. Thanks to the high spatial resolution of the focused ultrasound field, the focal point of the ultrasound can be positioned inside the soft actuator, which induces the limited surface temperature (i.e. contact with tissues. As shown in Supplementary Fig. 16) increase when compared with the higher temperature inside the actuator's body in actuated state and will not cause damage to surrounding tissues. The dynamic response ability to temperature changes and high spatial resolution of the focused ultrasound makes the FUPT-based soft actuator suitable for biomedical applications.”

2. A related question is how accurate can the authors control the temperature on the soft robots? Since the ultrasound wave can be partially reflected by bone or other intermediate medium, the energy delivered to the robots may not be the same in different scenarios. How to deal with this issue, especially for biomedical applications where slight temperature increase may have significant effect on the tissues.

Response:

The authors appreciate the reviewer's comments and agree that accurately controlling the temperature of the soft robot in complex internal environments can pose challenges. To address this issue, both open-loop and closed-loop control methods can be considered as potential strategies.

First, in an open-loop control approach, we can take into consideration of the environmental factors such as bone or other intermediate mediums by modeling their reflection and refraction of ultrasound waves. By calculating the energy of the ultrasound waves reaching the target location, we can roughly estimate the temperature of the soft robot, although this may be a complex task.

A more feasible and reliable approach is to employ a closed-loop control strategy. One established solution in clinical practice for real-time temperature monitoring is the use of magnetic resonance imaging (MRI)²³. Integrating MR thermometry would enable nearly real-time temperature monitoring and necessary adjustments to maintain precise temperature, which is applied in some commercial devices. Regarding accuracy, it have been reported that HIFU devices based on MR thermometry^{24,25} can achieve a temperature monitoring standard deviation of sub-degree level, some other study achieved a typical average temperature overshoot to 1 °C²⁶. Therefore, we believe that by combining with MR thermometry equipment, we can achieve accurate monitoring of the soft robot inside the body.

Additionally, our proposed ultrasound imaging system allows for real-time monitoring of the robot's inflation state. By measurement of its deformation, we can estimate the internal pressure. Although this method does not directly measure temperature, it provides a closed-loop approach that allows us to regulate the acoustic energy input to actuate the soft robot. By continuously monitoring the robot's inflation state, we can adjust the energy dosage to maintain the desired actuation state while minimizing the risks associated with overheating or over-pressurization, enhancing the overall safety of the actuation process.

We have added the discussion on **Page 23, line 392** shown as follows:

“It is worth noting that the proposed FUPT actuation method can seamlessly integrate with magnetic resonance thermometry, enabling precise and near real-time temperature control of the robot. This integration significantly expands the potential applications of FUPT-based medical soft robotics, opening up new possibilities for advanced medical procedures.”

3. In addition, soft materials are usually permeable to gas or even liquid. How to avoid the leakage of gas during heating? Will this cause safety issue for in vivo applications? How about the actuation performance of the soft robots in multiple cycles?

Response:

The authors appreciate the reviewer's comments. As the reviewer's suggestion, we conducted tests to evaluate the leakage of the internal material within the actuator. As shown in Fig. R3.4 (Supplementary Fig. 7), we injected approximately 1.1 g of Novec 7000 into an empty actuator and measured its mass after each test to estimate the leakage rate. Considering the application scenarios, we submerged the actuator in water at body temperature (i.e. $\sim 37\text{ }^{\circ}\text{C}$). As shown in Fig. R3.4a, the mass of the actuator gradually decreased over time, indicating the presence of internal material leakage. Within the 3 h testing duration, we observed a leakage of approximately 0.1 g (around 10% of the total phase transition liquid) with an average leakage rate of approximately 3.3% per hour. After approximately 16 h, the leakage of the enclosed liquid approximately 0.28 g ($\sim 25.5\%$), indicating a slower leakage rate compared to the initial state. It is important to note that our designed robot and actuators are intended for short-term operation rather than long-term implantation, so the amount of leakage within a relatively short period is negligible.

Fig. R3.4 (Supplementary Fig. 7): Leakage measurement of the phase-transition material inside the soft actuator over time. (a) In unactuated state ($37\text{ }^{\circ}\text{C}$) and (b) in actuated state ($45\text{ }^{\circ}\text{C}$).

For the actuator keeping in actuated state, the leakage rate may increase due to the increased vapor pressure. As shown in the red curve in Fig. R3.4b, the actuator was placed in the environment of 45 °C and keep in actuated state, the leakage rate increased to nearly triples (averaging 10% per hour for the first 3 hours) compared to the results measured in 37 °C. However, considering the short duration of each actuation cycle (as short as 2 minutes for the biopsy actuator tested in this experiment), the leakage due to heating is relatively small (about 0.1667% per minute in the actuated state) and can be negligible.

How to avoid the leakage of gas during heating?

To address the issue of gas leakage during heating, we propose selecting materials with better gas tightness, such as latex. As shown in Fig. R3.4, replacing the shell material from Ecoflex 00-30 (approximately 1 mm thick) to latex (approximately 0.1 mm thick) resulted in no significant leakage observed at both 37 °C and 45 °C (i.e. keeping in actuated state corresponding to the FUPT-based soft actuator). Therefore, for the FUPT-based soft robotics are to be extended for long-term implanted devices actuation, it would be necessary to replace or modify the materials to enhance their sealing properties.

Will this cause safety issue for in vivo applications?

As for the safety issue for in vivo applications, Novec 7000 has been found to have good biocompatibility. The lethal dose 50 (LD50) of Novec 7000 exceeds 2000 mg/kg, indicating a very low level of acute toxicity (in comparison, the LD50 of NaCl is about 3000 mg/kg). Furthermore, the HMIS Healthy Hazard Classification of Novec 7000 is 0²⁷, indicating a minimal health hazard. Considering the small amount of leakage, we believe there is negligible effect on the surrounding environment.

How about the actuation performance of the soft robots in multiple cycles?

In terms of the actuation performance of the soft robots in multiple cycles, we examined the influence of multiple actuation cycles on the leakage of the internal phase-change material using the same actuator. The results indicate a mass loss of 6.45% after 10 cycles and 12% after

20 cycles, which translates to an average leakage of approximately 7.2 mg of Novec 7000 per cycle. We acknowledge that the leakage does affect the performance of the actuator, with a decrease in volume expansion rate of approximately 10% after 20 cycles under the same actuation temperature conditions, resulting in a performance loss of approximately 0.5% per cycle. However, it is worth noting that similar actuation performance can be obtained after multiple actuation cycles by increasing the input ultrasonic energy to compensate for the performance loss due to leakage. As verified in Fig. R3.5 (Supplementary Fig. 8), where after 20 cycles, the actuator is still able to meet the design objective of extruding the internal component.

Fig. R3.5 (Supplementary Fig. 8): The repeatable actuation performance (i.e. 20 cycles) of the soft actuator.

We have revised the manuscript on **Page 23, line 396** shown as follows:

“While the FUPT method is demonstrated to be a noteworthy advance for the actuation of untethered soft robot, to ensure the adaptability of the FUPT-based soft robot for scenarios that involve a large number of actuation cycles or long-term implanted devices, we also propose further exploration to for lowering the vapor permeability. This could involve investigating alternative matrix that exhibit improved vapor sealing properties, as discussed in Supplementary Section 7.”

And we also added an external **Supplementary Section 7** to discuss the influence factors of enclosed liquid/air leakage as follows:

“Supplementary Section 7. Permeability evaluation of the phase transition material

To examine the leakage of the internal phase transition material of the actuator, we injected approximately 1.1 g of Novec 7000 into an empty actuator and measured its mass.....

Therefore, if the FUPT-based soft robotics are to be extended for long-term implanted devices actuation, it would be necessary to replace or modify the materials to enhance their sealing properties.” (Details can be found in Supplementary section 7 in the revised Supplementary information)

4. Can the authors compare the force (or force per gram/volume) and response time (or cooling time) of their robots with other approaches, such as embedding a shape memory alloy in soft elastomers? It’s better to add a figure or table to compare the performances.

Response:

The authors appreciate the reviewer’s suggestions and agree that a comparison table can further strengthen and highlight the advantages of this work. We have updated the performance comparison table to Supplementary information as Supplementary Table 3 (shown as Table R3.1 below).

As shown in Table R3.1, the soft robots based on the liquid-gas phase transition method (including FUPT) shows Newtons level force output (5.5 N for FUPT, up to 70 N for other works with same principle) that generally higher than others. During the comparison process, we recognized the significance of scale as an important factor. On the one hand, scale can serve as a reference along with the output force level to evaluate the mechanical performance of the driving principle. On the other hand, due to the scaling laws, thermally driven processes are expected to scale inversely with system size. This means that smaller structures tend to heat and cool more rapidly¹⁵, which can greatly impact the response time. However, while it is difficult for us to do the specific and fairness comparison due that the limited or without providing of the scale/weight information in the literature, we feel the Table R3.1 adequately

capture the advantages and biomedical application potentials of the proposed FUPT actuation method, due to the realizing of the combination of high spatial resolution, high tissue penetration depth and enough force output.

We have further edited the manuscript on **Supplementary Table 3** shown as follows:

Table R3.1 (Supplementary Table 3): The comparison of various actuation methods for untethered soft actuator and robot.

Strategy		Selective actuation	Newton-level force	Response time	Scale	Centimeter-level tissue penetration	Medical image	Ref
Quasistatic field	Magnetic soft robot	No	No (~ 60 μ N) ¹⁹	Much less than one second	Millimeter	Yes	Yes (Ultrasound)	8
		No		Much less than one second	Millimeter to centimeter	Yes	No	9
Electromagnetic wave	Liquid crystal elastomer	Yes (Spatial)	No	Less than one second	Submillimeter	No (~ 0.03-0.05 cm) ^{10,11}	No	12
		Yes (frequency)	No	A few seconds to tens of seconds	Millimeter to centimeter	No (~ 0.05-0.3 cm) ^{10,11}	No	13
	Laser-induced phase transition	Yes (Spatial)	N/A	Tens of seconds	Centimeter	No (~ 0.5 cm) ^{10,11}	No	14
	Millimeter wave-induced phase transition	Yes (Spatial)	Yes (38 N)	Ten seconds to tens of seconds	Centimeter	No (~ 0.05 cm) ¹⁵	No	16
	Shape memory alloy	Yes (frequency)	No (0.6 N)	A few seconds	Millimeter to centimeter	Yes	No	17
	Shape Memory Polymer	No	No	Ten seconds to tens of seconds	Millimeter to centimeter	Yes	No	18
	Coiled artificial muscle	No	Yes (~3.1 N)	A few seconds	Millimeter to centimeter	Yes	No	19
	Radio frequency-induced phase transition	No	Yes (3.1 N to 70 N)	A few seconds	Millimeter	Yes	Yes (Ultrasound)	20
No		Yes (up to 31 N)	Ten seconds to tens of seconds	Centimeter	Yes	No	21	
Mechanical wave	Focused ultrasound-controlled phase transition	Yes (Spatial)	Yes (up to 5.5 N)	A few seconds to tens of seconds	Millimeters to centimeter	Yes (~ 8 cm)	Yes (Ultrasound, X-ray)	This work

5. The reviewer wonders about the biocompatibility of the Novec 7000 engineering fluid.

Response:

We thank the reviewer's comment. Ensuring the safety of materials is indeed a crucial aspect for biomedical applications. The Novec 7000 has exceptional biocompatibility as supported by previous studies^{9,28}. The lethal dose 50 (LD50) of Novec 7000 exceeds 2000 mg/kg, indicating a very low level of acute toxicity (in comparison, the LD50 of NaCl is about 3000 mg/kg). Furthermore, the HMIS Healthy Hazard Classification of Novec 7000 is 0²⁷, indicating a minimal health hazard. Therefore, we believe these findings provide sufficient evidence of the biocompatibility profile of Novec 7000.

We have added the description of the biocompatibility on **Page 6, line 97** shown as follows:

“(i.e. Novec 7000 engineering fluid, which has a boiling point of 34 °C at 1 atm and exhibits excellent biocompatibility)³⁸”

6. Can the authors be more quantitative on selective actuation? Since there is thermal diffusion, the two actuated parts cannot be too close. What is the minimal distance between different actuation parts to guarantee selective actuation?

Response:

We thank the reviewer for making this point. In our theoretical analysis, the focal diameter of the ultrasound waves is approximately 3 mm as shown in Fig. 2b in the manuscript. Considering this, we have fabricated the multiunit soft capsule (i.e. four similar unit combination) with each unit measuring approximately 4 mm × 4 mm × 3.5 mm, to verify the independent control of two modules when their center-to-center distance is about 4.5 mm, as detailed in Fig. R3.6 (Fig. 4g and 4h). By aligning the focal point of the acoustic field with different units, selective release of one specific liquid cargo can be achieved (Fig. R3.6). The experimental results demonstrate that FUPT can achieve selective actuation of adjacent soft actuators with a central distance of approximately 4.5 mm.

In addition, we totally agree with the reviewer's suggestion that thermal diffusion can impact the minimal distance for selective actuation. Therefore, the materials with lower thermal conductivity were used between adjacent units to mitigate this effect (Fig. 4g and 4h). Which helps to reduce the influence of heat transfer to increase independent control ability.

Fig. R3.6 (Fig. 4) Multiunit soft capsule. (g) Structural design of the multiunit soft capsule. **(h)** A series of images demonstrating the selective liquid cargo release by aligning the high-intensity acoustic field with different chambers.

We have revised the **Fig. 4h** and the corresponding description on **Page 15, line 258** shown as follows:

“The experimental results demonstrate that FUPT can achieve selective actuation of adjacent soft actuators with a central distance of approximately 4.5 mm.”

References

1. Phung, D. C. *et al.* Combined hyperthermia and chemotherapy as a synergistic anticancer treatment. *J. Pharm. Investig.* **49**, 519–526 (2019).
2. Sofuni, A., Asai, Y., Mukai, S., Yamamoto, K. & Itoi, T. High-intensity focused ultrasound therapy for pancreatic cancer. *J. Med. Ultrason.* (2022).
3. Wu, F. *et al.* Tumor vessel destruction resulting from high-intensity focused ultrasound in patients with solid malignancies. *Ultrasound Med. Biol.* **28**, 535–542 (2002).
4. Jiménez-Gambín, S., Jiménez, N., Benlloch, J. M. & Camarena, F. Holograms to Focus Arbitrary Ultrasonic Fields through the Skull. *Phys. Rev. Appl.* **12**, 014016 (2019).
5. Jolesz, F. A. MRI-Guided Focused Ultrasound Surgery. *Annu. Rev. Med.* **60**, 417–430 (2009).
6. Li, S. *et al.* Preoperative T Staging of Advanced Gastric Cancer using Double Contrast-Enhanced Ultrasound. *Ultraschall Med.* **33**, E218–E224 (2012).
7. Andrzejewska, M. & Grzymisławski, M. The role of intestinal ultrasound in diagnostics of bowel diseases. *Przełąd Gastroenterol.* **13**, 1–5 (2018).
8. Moriwaki, T. *et al.* In Vitro Measurement of Contact Pressure Applied to a Model Vessel Wall during Balloon Dilatation by Using a Film-Type Sensor. *J. Neuroendovascular Ther.* **16**, 192–197 (2022).
9. Tang, Y. *et al.* Wireless Miniature Magnetic Phase-Change Soft Actuators. *Adv. Mater.* **34**, 2204185 (2022).
10. Li, M. *et al.* Miniature Coiled Artificial Muscle for Wireless Soft Medical Devices. *Sci. Adv.* **8**, eabm5616 (2022).
11. Hu, W., Lum, G. Z., Mastrangeli, M. & Sitti, M. Small-scale Soft-bodied Robot with Multimodal Locomotion. *Nature* **554**, 81–85 (2018).
12. Dong, Y. *et al.* Untethered Small-scale Magnetic Soft Robot with Programmable Magnetization and Integrated Multifunctional Modules. *Sci. Adv.* **8**, eabn8932 (2022).
13. Bashkatov, A. N., Genina, E. A., Kochubey, V. I. & Tuchin, V. V. Optical properties of human skin, subcutaneous and mucous tissues in the wavelength range from 400 to 2000 nm. *J. Phys. Appl. Phys.* **38**, 2543 (2005).
14. Avcı, P. *et al.* Low-level Laser (Light) Therapy (LLLT) in Skin: Stimulating, Healing, Restoring. *Semin. Cutan. Med. Surg.* **32**, 41–52 (2013).
15. Palagi, S. *et al.* Structured Light Enables Biomimetic Swimming and Versatile Locomotion of Photoresponsive Soft Microrobots. *Nat. Mater.* **15**, 647–653 (2016).
16. Zuo, B., Wang, M., Lin, B.-P. & Yang, H. Visible and Infrared Three-wavelength Modulated Multi-directional Actuators. *Nat. Commun.* **10**, 4539 (2019).
17. Meder, F., Naselli, G. A., Sadeghi, A. & Mazzolai, B. Remotely Light-Powered Soft Fluidic Actuators Based on Plasmonic-Driven Phase Transitions in Elastic Constraint. *Adv. Mater.* **31**, 1905671 (2019).
18. Wu, T., Rappaport, T. S. & Collins, C. M. The Human Body and Millimeter-wave Wireless Communication Systems: Interactions and Implications. in *2015 IEEE Int. Conf. Commun. (ICC)* 2423–2429 (2015).
19. Ueno, S. & Monnai, Y. Wireless Soft Actuator Based on Liquid-Gas Phase Transition Controlled by Millimeter-Wave Irradiation. *IEEE Robot. Autom. Lett.* **5**, 6483–6488 (2020).
20. Boyvat, M., Koh, J.-S. & Wood, R. J. Addressable wireless actuation for multijoint folding robots and devices. *Sci. Robot.* **2**, eaan1544 (2017).
21. Ze, Q. *et al.* Magnetic Shape Memory Polymers with Integrated Multifunctional Shape Manipulation. *Adv. Mater.* **32**, 1906657 (2020).
22. Mirvakili, S. M., Sim, D., Hunter, I. W. & Langer, R. Actuation of Untethered Pneumatic Artificial Muscles and Soft Robots Using Magnetically Induced Liquid-to-gas Phase Transitions. *Sci. Robot.* **5**, eaaz4239 (2020).
23. Rieke, V. & Pauly, K. B. MR Thermometry. *J. Magn. Reson. Imaging JMRI* **27**, 376–390 (2008).
24. Holbrook, A. B., Santos, J. M., Kaye, E., Rieke, V. & Pauly, K. B. Real Time MR Thermometry for Monitoring HIFU Ablations of the Liver. *Magn. Reson. Med.* **63**, 365–373

(2010).

25. Staruch, R., Chopra, R. & Hynynen, K. Hyperthermia in Bone Generated with MR Imaging-controlled Focused Ultrasound: Control Strategies and Drug Delivery. *Radiology* **263**, 117–127 (2012).

26. Mougnot, C. *et al.* Three-dimensional spatial and temporal temperature control with MR thermometry-guided focused ultrasound (MRgHIFU). *Magn. Reson. Med.* **61**, 603–614 (2009).

27. 3M Novec 7000, safety data sheet.

https://multimedia.3m.com/mws/mediawebserver?mwsId=SSSSSuUn_zu8l00x482vOxme4v70k17zHvu9lxtD7SSSSSS--.

28. Hirai, S. *et al.* Micro Elastic Pouch Motors: Elastically Deformable and Miniaturized Soft Actuators Using Liquid-to-Gas Phase Change. *IEEE Robot. Autom. Lett.* **6**, 5373–5380 (2021).

REVIEWERS' COMMENTS

Reviewer #1 (Remarks to the Author):

Thank you, my concerns have been addressed.

Reviewer #2 (Remarks to the Author):

This paper presents a focused ultrasound-controlled phase transition (FUPT) strategy for achieving millimetre-level spatially selective actuation and Newton-level force of soft robots operating within tissue or liquid environments. A soft robot for on-demand liquid cargo delivery in the pipe and a biopsy robot for tissue acquisition and patching in the intestine are designed to verify their application potential. A control system combining the FUPT actuation method and ultrasound imaging is demonstrated to achieve precise operations. The authors have made the following modifications to the previous revision comments :

1. The authors evaluated its stability and temperature sensitivity in simulated human temperature environment. The actuator can maintain a stable state in the human body temperature condition, while also achieving actuation when stimulated by focused ultrasound. The supplementary experimental data and analysis have solved my previous confusion.
2. The authors provided a detailed answer to question 2. The authors clarified that the Von Mises stress depicted in Fig. 3b and 3d represents the maximum tension within the soft material of the actuator, which is much higher than the contact pressure with tissues, and gave a schematic of the force relationship of internal pressure, the tension in the actuator's skin (Von miss stress) and contact pressure. The author also emphasized the meaning of maximum Von Mises stress in the corresponding paragraphs.
3. The authors conducted experiments to investigate the anchor performance of the expansion actuator in the presence of mucus in a biological pipeline (i.e. esophagus). The experimental results indicate that even in soft biological pipelines with the presence of mucus, the robot still possesses anchoring ability, which provide a good explanation for question 3.
4. The author provided a detailed explanation of question 4. The author acknowledged that the motion of robots was indeed influenced by high Reynolds number environments. However, the robot designed by the authors was mainly used for scenarios with limited fluid flow, such as the gastrointestinal tract. Therefore, the robot rarely encountered such situations.
5. According to the review comments, the author has compiled a very detailed table comparing the FUPT method proposed in this article with other actuation strategies in different aspects. Table R2.1 well reflected the characteristics of the FUPT method proposed in this article.

Overall, the author has made detailed revisions to the review comments, and I suggest considering accepting this article.

Reviewer #3 (Remarks to the Author):

The authors have added comprehensive discussions and experiments to address my previous concerns. I would like to thank the authors for providing detailed, well-organized responses. I enjoyed reading the revised manuscript and recommend publication.

Response to referees

Reviewer #1:

Thank you, my concerns have been addressed.

Response:

We express our sincere appreciation to the reviewer for their kind comments and invaluable feedback, which have significantly enhanced the quality of our research.

Reviewer #2:

This paper presents a focused ultrasound-controlled phase transition (FUPT) strategy for achieving millimetre-level spatially selective actuation and Newton-level force of soft robots operating within tissue or liquid environments. A soft robot for on-demand liquid cargo delivery in the pipe and a biopsy robot for tissue acquisition and patching in the intestine are designed to verify their application potential. A control system combining the FUPT actuation method and ultrasound imaging is demonstrated to achieve precise operations. The authors have made the following modifications to the previous revision comments:

1. The authors evaluated its stability and temperature sensitivity in simulated human temperature environment. The actuator can maintain a stable state in the human body temperature condition, while also achieving actuation when stimulated by focused ultrasound. The supplementary experimental data and analysis have solved my previous confusion.
2. The authors provided a detailed answer to question 2. The authors clarified that the Von Mises stress depicted in Fig. 3b and 3d represents the maximum tension within the soft material of the actuator, which is much higher than the contact pressure with tissues, and gave a schematic of the force relationship of internal pressure, the tension in the actuator's skin (Von miss stress) and contact pressure. The author also emphasized the meaning of maximum Von Mises stress in the corresponding paragraphs.
3. The authors conducted experiments to investigate the anchor performance of the expansion actuator in the presence of mucus in a biological pipeline (i.e. esophagus). The experimental results indicate that even in soft biological pipelines with the presence of mucus, the robot still possesses anchoring ability, which provide a good explanation for question 3.
4. The author provided a detailed explanation of question 4. The author acknowledged that the motion of robots was indeed influenced by high Reynolds number environments. However, the robot designed by the authors was mainly used for scenarios with limited fluid flow, such as the gastrointestinal tract. Therefore, the robot rarely encountered such situations.
5. According to the review comments, the author has compiled a very detailed table comparing the FUPT method proposed in this article with other actuation strategies in different aspects. Table R2.1 well reflected the characteristics of the FUPT method proposed in this article.

Overall, the author has made detailed revisions to the review comments, and I suggest considering accepting this article.

Response:

We express our sincere appreciation to the reviewer for their kind comments and invaluable feedback, which have significantly enhanced the quality of our research.

Reviewer #3:

The authors have added comprehensive discussions and experiments to address my previous concerns. I would like to thank the authors for providing detailed, well-organized responses. I enjoyed reading the revised manuscript and recommend publication.

Response:

We express our sincere appreciation to the reviewer for their kind comments and invaluable feedback, which have significantly enhanced the quality of our research.